# Sheaf Hypergraph Networks

**Iulia Duta**
University of Cambridge
id366@cam.ac.uk

**Giulia Cassarà**
University of Rome, La Sapienza
giulia.cassara@uniroma1.it

**Fabrizio Silvestri**
University of Rome, La Sapienza
fabrizio.silvestri@uniroma1.it

**Pietro Liò**
University of Cambridge
pl219@cam.ac.uk

## Abstract

Higher-order relations are widespread in nature, with numerous phenomena involving complex interactions that extend beyond simple pairwise connections. As a result, advancements in higher-order processing can accelerate the growth of various fields requiring structured data. Current approaches typically represent these interactions using hypergraphs. We enhance this representation by introducing cellular sheaves for hypergraphs, a mathematical construction that adds extra structure to the conventional hypergraph while maintaining their local, higher-order connectivity. Drawing inspiration from existing Laplacians in the literature, we develop two unique formulations of sheaf hypergraph Laplacians: linear and non-linear. Our theoretical analysis demonstrates that incorporating sheaves into the hypergraph Laplacian provides a more expressive inductive bias than standard hypergraph diffusion, creating a powerful instrument for effectively modelling complex data structures. We employ these sheaf hypergraph Laplacians to design two categories of models: Sheaf Hypergraph Neural Networks and Sheaf Hypergraph Convolutional Networks. These models generalize classical Hypergraph Networks often found in the literature. Through extensive experimentation, we show that this generalization significantly improves performance, achieving top results on multiple benchmark datasets for hypergraph node classification.

## 1 Introduction

The prevalence of relational data in real-world scenarios has led to rapid development and widespread adoption of graph-based methods in numerous domains [1–4]. However, a major limitation of graphs is their inability to represent interactions that goes beyond pairwise relations. In contrast, real-world interactions are often complex and multifaceted. There is evidence that higher-order relations frequently occur in neuroscience [5, 6], chemistry [7], environmental science [8, 9] and social networks [10]. Consequently, learning powerful and meaningful representations for hypergraphs has emerged as a promising and rapidly growing subfield of deep learning [11–16]. However, current hypergraph-based models struggle to capture higher-order relationships effectively. As described in [17], conventional hypergraph neural networks often suffer from the problem of over-smoothing. As we propagate the information inside the hypergraph, the representations of the nodes become uniform across neighbourhoods. This effect hampers the capability of hypergraph models to capture local, higher-order nuances.

More powerful and flexible mathematical constructs are required to capture real-world interactions' complexity better. Sheaves provide a suitable enhancement for graphs that allow for more diverse and expressive representations. A cellular sheaf [18] enables to attach data to a graph, by associating vector spaces to the nodes, together with a mechanism of transferring the information along the

37th Conference on Neural Information Processing Systems (NeurIPS 2023).

edges. This approach allows for richer data representation and enhances the ability to model complex interactions.

Motivated by the need for more expressive structures, we introduce a *cellular sheaf for hypergraphs*, which allows for the representation of more sophisticated dynamics while preserving the higher-order connectivity inherent to hypergraphs. We take on the non-trivial challenge to generalize the two commonly used hypergraph Laplacians [19, 11] to incorporate the richer structure sheaves offer. Theoretically, we demonstrate that the diffusion process derived using the *sheaf hypergraph Laplacians* that we propose induces a more expressive inductive bias than the classical hypergraph diffusion. Leveraging this enhanced inductive bias, we construct and test two powerful neural networks capable of inferring and processing hypergraph sheaf structure: the *Sheaf Hypergraph Neural Network* (SheafHyperGNN) and the *Sheaf Hypergraph Convolutional Network* (SheafHyperGCN).

The introduction of the cellular sheaf for hypergraphs expands the potential for representing complex interactions and provides a foundation for more advanced techniques. By generalizing the hypergraph Laplacians with the sheaf structure, we can better capture the nuance and intricacy of real-world data. Furthermore, our theoretical analysis provides evidence that the sheaf hypergraph Laplacians embody a more expressive inductive bias, essential for obtaining strong representations.

**Our main contributions** are summarised as follow:

1. We introduce the **cellular sheaf for hypergraphs**, a mathematical construct that enhances the hypergraphs with additional structure by associating a vector space with each node and hyperedge, along with linear projections that enable information transfer between them.

2. We propose both a **linear** and a **non-linear sheaf hypergraph Laplacian**, generalizing the standard hypergraph Laplacians commonly used in the literature. We also provide a theoretical characterization of the inductive biases generated by the diffusion processes of these Laplacians, showcasing the benefits of utilizing these novel tools for effectively modeling intricate phenomena.

3. The two sheaf hypergraph Laplacians are the foundation for **two novel architectures** tailored for hypergraph processing: **Sheaf Hypergraph Neural Network** and **Sheaf Hypergraph Convolutional Network**. Experimental findings demonstrate that these models achieve top results, surpassing existing methods on numerous benchmarking datasets.

## 2   Related work

**Sheaves on Graphs.** Utilizing graph structure in real-world data has improved various domains like healthcare [1], biochemistry [2], social networks [20], recommendation systems [3], traffic prediction [21], with graph neural networks (GNNs) becoming the standard for graph representations. However, in heterophilic setups, when nodes with different labels are likely to be connected, directly processing the graph structure leads to weak performance. In [22], they address this by attaching additional geometric structure to the graph, in the form of cellular sheaves [18].

A cellular sheaf on graphs associates a vector space with each node and each edge together with a linear projection between these spaces for each incident pair. To take into account this more complex geometric structure, SheafNN [23] generalised the classical GNNs [24–26] by replacing the graph Laplacian with a sheaf Laplacian [27]. Higher-dimensional sheaf-based neural networks are explored, with sheaves either learned from the graph [22] or deterministically inferred for efficiency [28]. Recent methods integrate attention mechanisms [29] or replace propagation with wave equations [30]. In recent developments, Sheaf Neural Networks have been found to significantly enhance the performance of recommendation systems, as they improve upon the limitations of graph neural networks [31].

In the domain of heterogeneous graphs, the concept of learning unique message functions for varying edges is well-established. However, there's a distinction in how sheaf-based methods approach this task compared to heterogeneous methods such as RGCN [32]. Unlike the latter, which learns individual parameters for each kind of incident relationship, sheaf-based methods dynamically predict projections for each relationship, relying on features associated with the node and hyperedge. As a result, the total parameters in sheaf networks do not escalate with an increase in the number of hyperedges. This difference underscores a fundamental shift in paradigm between the two methods.

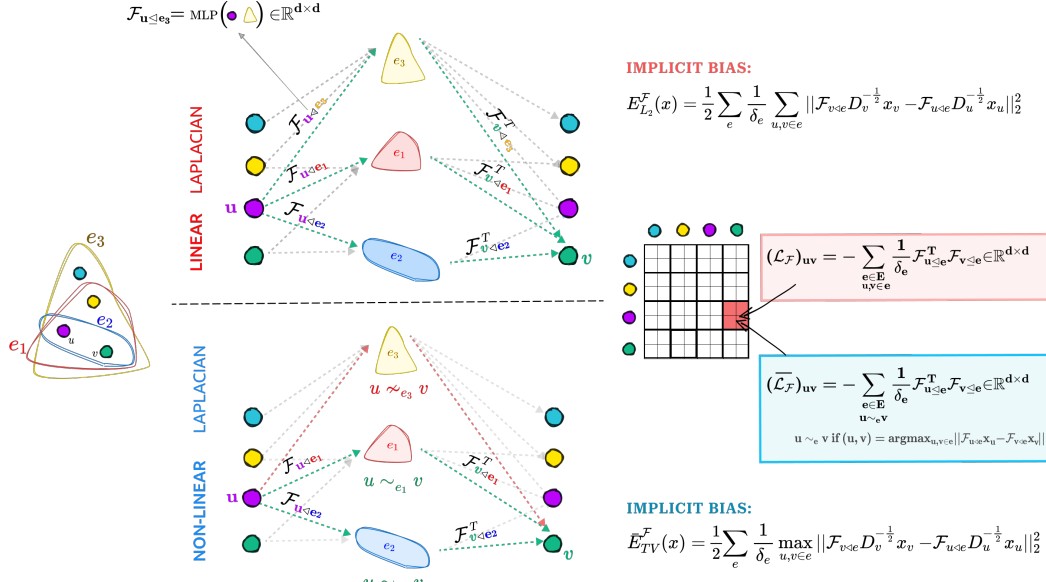

Figure 1: Visual representation of linear and non-linear sheaf hypergraph Laplacian. (**Top**) In the linear case, the block matrix $(\mathcal{L}_\mathcal{F})_{uv}$ corresponding to the pair of nodes $(u, v)$ accumulates contributions from each hyperedge that simultaneously contains both nodes. (**Bottom**) In the non-linear version, for each hyperedge, we first select the two nodes that are the most dissimilar in the hyperedge stalk domain: $u \sim_e v$ if $(u, v) = argmax_{u,v \in e} ||\mathcal{F}_{u \lhd e} x_u - \mathcal{F}_{v \lhd e} x_v||_2^2$. Then, the block matrix $(\bar{\mathcal{L}}_\mathcal{F})_{uv}$ associated with the pair of nodes $(u, v)$ only accumulates contributions from a hyperedge $e$ if $u \sim_e v$. The two operators (linear and non-linear sheaf hypergraph Laplacian) represent the building blocks for the Sheaf Hypergraph Neural Network and Sheaf Hypergraph Convolutional Network respectively and we theoretically show that they exhibit a more expressive implicit bias compared to the traditional Hypergraph Networks, leading to better performance.

**Hypergraph Networks.** Graphs, while useful, have a strong limitation: they represent only pairwise relations. Many natural phenomena involve complex, higher-order interactions [33–35, 9], requiring a more general structure like hypergraphs. Recent deep learning approaches have been developed for hypergraph structures. HyperGNN [11] expands the hypergraph into a weighted clique and applies message passing similar to GCNs [24]. HNHN [36] improves this with non-linearities, while HyperGCN [37] connects only the most discrepant nodes using a non-linear Laplacian. Similar to the trend in GNNs, attention models gain popularity also in the hypergraph domain. HCHA [38] uses an attention-based incidence matrix, computed based on a nodes-hyperedge similarity. Similarly, HERALD [39] uses a learnable distance to infer a soft incidence matrix. On the other hand, HEAT [15] creates messages by propagating information inside each hyperedge using Transformers [40].

Many hypergraph neural network (HNN) methods can be viewed as two-stage frameworks: 1) sending messages from nodes to hyperedges and 2) sending messages back from hyperedges to nodes. Thus, [41] proposes a general framework where the first step is the average operator, while the second stage could use any existing GNN module. Similarly, [42] uses either DeepSet functions [43] or Transformers [40] to implement the two stages, while [44] uses a GNN-like aggregator in both stages, with distinct messages for each (node, hyperedge) pair.

In contrast, we propose a novel model to improve the hypergraph processing by attaching a cellular sheaf to the hypergraph structure and diffusing the information inside the model according to it. We will first introduce the cellular sheaf for hypergraph, prove some properties for the associated Laplacians, and then propose and evaluate two architectures based on the sheaf hypergraph Laplacians.

## 3 Hypergraph Sheaf Laplacian

An undirected hypergraph is a tuple $\mathcal{H} = (V, E)$ where $V = \{1, 2 \ldots n\}$ is a set of nodes (also called vertices), and $E$ is a set of hyperedges (also called edges when there is no confusion with the graph

edges). Each hyperedge $e$ is a subset of the nodes set $V$. We denote by $n = |V|$ the number of nodes in the hypergraph $\mathcal{H}$ and by $m = |E|$ the number of hyperedges. In contrast to graph structures, where each edge contains exactly two nodes, in a hypergraph an edge $e$ can contain any number of nodes. The number of nodes in each hyperedge ($|e|$) is called the *degree of the hyperedge* and is denoted by $\delta_e$. In contrast, the number of hyperedges containing each node $v$ is called the *degree of the node* and is denoted by $d_v$.

Following the same intuition from defining sheaves on graphs [23, 22], we will introduce the cellular sheaf associated with a hypergraph $\mathcal{H}$.

**Definition 1.** A *cellular sheaf $\mathcal{F}$ associated with a hypergraph $\mathcal{H}$* is defined as a triple $\langle \mathcal{F}(v), \mathcal{F}(e), \mathcal{F}_{v \trianglelefteq e} \rangle$, where:

1. $\mathcal{F}(v)$ are *vertex stalks:* vector spaces associated with each node $v$;
2. $\mathcal{F}(e)$ are *hyperedge stalks:* vector spaces associated with each hyperedge $e$;
3. $\mathcal{F}_{v \trianglelefteq e} : \mathcal{F}(v) \to \mathcal{F}(e)$ are *restriction maps:* linear maps between each pair $v \trianglelefteq e$, if hyperedge $e$ contains node $v$.

In simpler terms, a sheaf associates a space with each node and each hyperedge in a hypergraph and also provides a linear projection that enables the movement of representations between nodes and hyperedges, as long as they are adjacent. Unless otherwise specified, we assign the same d-dimensional space for all vertex stalks $\mathcal{F}(v) = \mathbb{R}^d$ and all hyperedge stalks $\mathcal{F}(e) = \mathbb{R}^d$. We refer to $d$ as the dimension of the sheaf.

Previous works focused on creating hypergraph representations by relying on various methods of defining a Laplacian for a hypergraph. In this work, we will concentrate on two definitions: a linear version of the hypergraph Laplacian as used in [11], and a non-linear version of the hypergraph Laplacian as in [37]. We will extend both of these definitions to incorporate the hypergraph sheaf structure, analyze the advantages that arise from this, and propose two different neural network architectures based on each one of them. For a visual comparison between the two proposed sheaf hypergraph Laplacians, see Figure 1.

## 3.1   Linear Sheaf Hypergraph Laplacian

**Definition 2.** Following the definition of a cellular sheaf on hypergraphs, we introduce the *linear sheaf hypergraph Laplacian* associated with a hypergraph $\mathcal{H}$ as $(\mathcal{L}_\mathcal{F})_{vv} = \sum\limits_{e; v \in e} \frac{1}{\delta_e} \mathcal{F}_{v \trianglelefteq e}^T \mathcal{F}_{v \trianglelefteq e} \in \mathbb{R}^{d \times d}$ and $(\mathcal{L}_\mathcal{F})_{uv} = - \sum\limits_{e; u, v \in e} \frac{1}{\delta_e} \mathcal{F}_{u \trianglelefteq e}^T \mathcal{F}_{v \trianglelefteq e} \in \mathbb{R}^{d \times d}$, where $\mathcal{F}_{v \trianglelefteq e} : \mathbb{R}^d \to \mathbb{R}^d$ represents the linear restriction maps guiding the flow of information from node $v$ to hyperedge $e$.

The linear sheaf Laplacian operator for node $v$ applied on a signal $x \in \mathbb{R}^{n \times d}$ can be rewritten as:

$$\mathcal{L}_\mathcal{F}(x)_v = \sum_{e; v \in e} \frac{1}{\delta_e} \mathcal{F}_{v \trianglelefteq e}^T \left( \sum_{\substack{u \in e \\ u \neq v}} (\mathcal{F}_{v \trianglelefteq e} x_v - \mathcal{F}_{u \trianglelefteq e} x_u) \right). \tag{1}$$

When each hyperedge contains exactly two nodes (thus $\mathcal{H}$ is a graph), the internal summation will contain a single term, and we recover the sheaf Laplacian for graphs as formulated in [22].

On the other hand, for the trivial sheaf, when the vertex and hyperedge stalks are both fixed to be $\mathbb{R}$ and the restriction map is the identity $\mathcal{F}_{v \trianglelefteq e} = 1$ we recover the usual linear hypergraph Laplacian [11, 45] defined as $\mathcal{L}(x)_v = \sum_{e; v \in e} \frac{1}{\delta_e} \sum_{u \in e} (x_v - x_u)$. However, when we allow for higher-dimensional stalks $\mathbb{R}^d$, the restriction maps for each adjacency pair $(v, e)$ become linear projections $\mathcal{F}_{v \trianglelefteq e} \in \mathbb{R}^{d \times d}$, enabling us to model more complex propagations, customized for each incident (node, hyperedge) pairs.

In the following sections, we will demonstrate the advantages of using this sheaf hypergraph diffusion instead of the usual hypergraph diffusion.

**Reducing energy via linear diffusion.** Previous work [17] demonstrates that diffusion using the classical symmetric normalised version of the hypergraph Laplacian $\Delta = D^{-\frac{1}{2}} \mathcal{L} D^{-\frac{1}{2}}$, where $D$ is a diagonal matrix containing the degrees of the vertices, reduces the following energy function:

$E_{L_2}(x) = \frac{1}{2} \sum_e \frac{1}{\delta_e} \sum_{u,v \in e} ||d_v^{-\frac{1}{2}} x_v - d_u^{-\frac{1}{2}} x_u||_2^2$. Intuitively, this means that applying diffusion using the *linear hypergraph Laplacian* leads to similar representations for neighbouring nodes. While this is desirable in some scenarios, it may cause poor performance in others, a phenomenon known as over-smoothing [17]. In the following, we show that applying diffusion using the linear *sheaf* hypergraph Laplacian addresses these limitations by implicitly minimizing a more expressive energy function. This allows us to model phenomena that were not accessible using the usual Laplacian.

**Definition 3.** We define *sheaf Dirichlet energy* of a signal $x \in \mathbb{R}^{n \times d}$ *on a hypergraph* $\mathcal{H}$ as:

$$E_{L_2}^{\mathcal{F}}(x) = \frac{1}{2} \sum_e \frac{1}{\delta_e} \sum_{u,v \in e} || \boxed{\mathcal{F}_{v \trianglelefteq e}} D_v^{-\frac{1}{2}} x_v - \boxed{\mathcal{F}_{u \trianglelefteq e}} D_u^{-\frac{1}{2}} x_u||_2^2,$$

where $D_v = \sum_{e; v \in e} \mathcal{F}_{v \trianglelefteq e}^T \mathcal{F}_{v \trianglelefteq e}$ is a normalisation term equivalent to the nodes degree $d_v$ for the trivial sheaf and $D = diag(D_1, D_2 \ldots D_n)$ the corresponding block diagonal matrix.

This quantity measures the discrepancy between neighbouring nodes in the hyperedge stalk domain as opposed to the usual Dirichlet energy for hypergraphs that, instead, measures this distance in the node features domain. In the following, we are showing that, applying hypergraph diffusion using the linear sheaf Laplacian implicitly reduces this energy.

**Proposition 1.** *The diffusion process using a symmetric normalised version of the linear sheaf hypergraph Laplacian minimizes the sheaf Dirichlet energy of a signal $x$ on a hypergraph $\mathcal{H}$. Moreover, the energy decreases with each layer of diffusion.*

Concretely, defining the diffusion process as $Y = (I - \Delta^{\mathcal{F}})X$ where $\Delta^{\mathcal{F}} = D^{-\frac{1}{2}} \mathcal{L}^{\mathcal{F}} D^{-\frac{1}{2}} \in \mathbb{R}^{nd \times nd}$ represents the symmetric normalised version of the linear sheaf hypergraph Laplacian, we have that $E_{L_2}^{\mathcal{F}}(Y) < \lambda_* E_{L_2}^{\mathcal{F}}(X)$, with $\{\lambda_i\}$ the non-zero eigenvalues of $\Delta^{\mathcal{F}}$ and $\lambda_* = \max_i \{(1 - \lambda_i)^2\} < 1$. All the proofs are in the Supplementary Material.

This result addresses some of the limitations of standard hypergraph processing. First, while classical diffusion using hypergraph Laplacian brings closer representations of the nodes in the nodes space $(x_v, x_u)$, our linear sheaf hypergraph Laplacian allows us to bring closer representations of the nodes in the more complex space associated with the hyperedges $(\mathcal{F}_{v \trianglelefteq e} x_v, \mathcal{F}_{u \trianglelefteq e} x_u)$. This encourages a form of hyperedge agreement, while preventing the nodes to become uniform. Secondly, in the hyperedge stalks, each node can have a different representation for each hyperedge it is part of, leading to a more expressive processing compared to the classical methods. Moreover, in many Hypergraph Networks, the hyperedges uniformly aggregate information from all its components. Through the presence of a restriction map for each (node, hyperedge) pair, we enable the model to learn the individual contribution that each node sends to each hyperedge.

From an opinion dynamic perspective [46] when the hyperedges represent group discussions, the input space $x_v$ can be seen as the private opinion, while the hyperedge stalk $\mathcal{F}_{v \trianglelefteq e} x_v$ can be seen as a public opinion (what an individual $v$ decide to express in a certain group $e$). Minimizing the *Dirichlet energy* creates private opinions that are in consensus inside each hyperedge, while minimizing the *sheaf Dirichlet energy* creates an *apparent* consensus, by only uniformizing the expressed opinion. Through our sheaf setup, each individual is allowed to express varying opinion in each group it is part of, potentially different than their personal belief. This introduces a realistic scenario inaccessible in the original hypergraph diffusion setup.

### 3.2 Non-Linear Sheaf Hypergraph Laplacian

Although the linear hypergraph Laplacian is commonly used to process hypergraphs, it falls short in fully preserving the hypergraph structure [47]. To address these shortcomings, [48] introduces the non-linear Laplacian, demonstrating that its spectral properties are more suited for higher-order processing compared to the linear Laplacian. For instance, compared to the linear version, the non-linear Laplacian leads to a more balanced partition in the minimum cut problem, a task known to be tightly related to the semi-supervised node classification. Additionally, while the linear Laplacian associates a clique for each hyperedge, the non-linear one offers the advantage of relying on a much sparser connectivity. We will adopt a similar methodology to derive the non-linear version of the sheaf hypergraph Laplacian and analyze the benefits of applying diffusion using this operator.

**Definition 4.** We introduce the *non-linear sheaf hypergraph Laplacian* of a hypergraph $\mathcal{H}$ with respect to a signal $x$ as following:

1. For each hyperedge $e$, compute $(u_e, v_e) = argmax_{u,v \in e}||\mathcal{F}_{u \triangleleft e}x_u - \mathcal{F}_{v \triangleleft e}x_v||$, the set of pairs containing the nodes with the most discrepant features in the hyperedge stalk.

2. Build an undirected graph $\mathcal{G}_H$ containing the same sets of nodes as $\mathcal{H}$ and, for each hyperedge $e$ connects the most discrepant nodes $(u, v)$ (from now on we will write $u \sim_e v$ if they are connected in the $\mathcal{G}_H$ graph due to the hyperedge e). If multiple pairs have the same maximum discrepancy, we will randomly choose one of them.

3. Define the sheaf non-linear hypergraph Laplacian as:

$$\bar{\mathcal{L}}_\mathcal{F}(x)_v = \sum_{e; u \sim_e v} \frac{1}{\delta_e} \mathcal{F}_{v \trianglelefteq e}^T (\mathcal{F}_{v \trianglelefteq e}x_v - \mathcal{F}_{u \trianglelefteq e}x_u). \tag{2}$$

Note that the sheaf structure impacts the non-linear diffusion in two ways: by shaping the graph structure creation (Step 1), where the two nodes with the greatest distance in the hypergraph stalk are selected rather than those in the input space; and by influencing the information propagation process (Step 3). When the sheaf is restricted to the trivial case ($d = 1$ and $\mathcal{F}_{v \trianglelefteq e} = 1$) this corresponds to the non-linear sheaf Laplacian of a hypergraph as introduced in [48].

**Reducing total variation via non-linear diffusion.** In the following discussion, we will demonstrate how transitioning from a linear to a non-linear sheaf hypergraph Laplacian alters the energy guiding the inductive bias. This phenomenon was previously investigated for the classical hypergraph Laplacian, with [48] revealing enhanced expressivity in the non-linear case.

**Definition 5.** We define *the sheaf total variation* of a signal $x \in \mathbb{R}^{n \times d}$ on a hypergraph $\mathcal{H}$ as:

$$\bar{E}_{TV}^\mathcal{F}(x) = \frac{1}{2} \sum_e \frac{1}{\delta_e} \max_{u,v \in e} || \mathcal{F}_{v \trianglelefteq e} D_v^{-\frac{1}{2}} x_v - \mathcal{F}_{u \trianglelefteq e} D_u^{-\frac{1}{2}} x_u ||_2^2,$$

where $D_v = \sum_{e; v \in e} \mathcal{F}_{v \trianglelefteq e}^T \mathcal{F}_{v \trianglelefteq e}$ is a normalisation term equivalent to the node's degree in the classical setup and $D = diag(D_1, D_2 \ldots D_n)$ is the corresponding block diagonal matrix.

This quantity generalised the total variance (TV) $\bar{E}_{TV}(x) = \frac{1}{2} \sum_e \frac{1}{\delta_e} \max_{u,v \in e} ||d_v^{-\frac{1}{2}} x_v - d_u^{-\frac{1}{2}} x_u||_2^2$ minimized in the non-linear hypergraph label propagation [48, 49]. Unlike the TV, the sheaf total variation measures the highest discrepancy at the hyperedge level computed in the hyperedge stalk, as opposed to the TV, which gauges the highest discrepancy in the feature space. We will explore the connection between the sheaf TV and our *non-linear sheaf hypergraph diffusion*.

**Proposition 2.** *The diffusion process using the symmetric normalised version of non-linear sheaf hypergraph Laplacian minimizes the sheaf total variance of a signal $x$ on hypergraph $\mathcal{H}$.*

Despite the change in the potential function being minimized, the overarching objective remains akin to that of the linear case: striving to achieve a coherent consensus among the representations within the hyperedge stalk space, rather than generating uniform features for each hyperedge in the input space. In contrast to the linear scenario, where a quadratic number of edges is required for each hyperedge, the non-linear sheaf hypergraph Laplacian associates a single edge with each hyperedge, thereby enhancing computational efficiency.

### 3.3 Sheaf Hypergraph Networks

Popular hypergraph neural networks [45, 11, 37, 50] draw inspiration from a variety of hypergraph diffusion operators [47, 48, 51], giving rise to diverse message passing techniques. These techniques all involve the propagation of information from nodes to hyperedges and vice-versa. We will adopt a similar strategy and introduce the Sheaf Hypergraph Neural Network and Sheaf Hypergraph Convolutional Network, based on two message-passing schemes inspired by the sheaf diffusion mechanisms discussed in this paper.

Given a hypergraph $\mathcal{H} = (V, E)$ with nodes characterised by a set of features $X \in \mathbb{R}^{n \times f}$, we initially linearly project the input features into $\tilde{X} \in \mathbb{R}^{n \times (df)}$ and then reshape them into $\tilde{X} \in \mathbb{R}^{nd \times f}$. As a result, each node is represented in the vertex stalk as a matrix $\mathbb{R}^{d \times f}$, where $d$ denotes the dimension of the vertex stalk, and $f$ indicates the number of channels.

Table 1: **Performance on a collection of hypergraph benchmarks.** Our models using sheaf hypergraph Laplacians demonstrate a clear advantage over their counterparts using classical Laplacians (HyperGNN and HyperGCN). Compared to other recent methods, SheafHyperGNN and SheafHyperGCN achieve competitive performance and attain state-of-the-art results in five of the datasets.

| Name | Cora | Citeseer | Pubmed | Cora_CA | DBLP_CA | Senate | House | Congress |
|---|---|---|---|---|---|---|---|---|
| HCHA | 79.14 ±1.02 | 72.42 ±1.42 | 86.41 ±0.36 | 82.55 ±0.97 | 90.92 ±0.22 | 48.62 ±4.41 | 61.36 ±2.53 | 90.43 ±1.20 |
| HNHN | 76.36 ±1.92 | 72.64 ±1.57 | 86.90 ±0.30 | 77.19 ±1.49 | 86.78 ±0.29 | 50.93 ±6.33 | 67.8 ±2.59 | 53.35 ±1.45 |
| AllDeepSets | 76.88 ±1.80 | 70.83 ±1.63 | 88.75 ±0.33 | 81.97 ±1.50 | 91.27 ±0.27 | 48.17 ±5.67 | 67.82 ±2.40 | 91.80 ±1.53 |
| AllSetTransformers | 78.58 ±1.47 | 73.08 ±1.20 | 88.72 ±0.37 | 83.63 ±1.47 | 91.53 ±0.23 | 51.83 ±5.22 | 69.33 ±2.20 | 92.16 ±1.05 |
| UniGCNII | 78.81 ±1.05 | 73.05 ±2.21 | 88.25 ±0.33 | 83.60 ±1.14 | 91.69 ±0.19 | 49.30 ±4.25 | 67.25 ±2.57 | 94.81 ±0.81 |
| HyperND | 79.20 ±1.14 | 72.62 ±1.49 | 86.68 ±0.43 | 80.62 ±1.32 | 90.35 ±0.26 | 52.82 ±3.20 | 51.70 ±3.37 | 74.63 ±3.62 |
| ED-HNN | 80.31 ±1.35 | 73.70 ±1.38 | **89.03 ±0.53** | 83.97 ±1.55 | **91.90 ±0.19** | 64.79 ±5.14 | 72.45 ±2.28 | **95.00 ±0.99** |
| HyperGCN[1] | 78.36 ±2.01 | 71.01 ±2.21 | 80.81 ±12.4 | 79.50 ±2.11 | 89.42 ±0.16* | 51.13 ±4.15 | 69.29 ±2.05 | 89.67 ±1.22 |
| SheafHyperGCN | 80.06 ±1.12 | 73.27 ±0.50 | 87.09 ±0.71 | 83.26 ±1.20 | 90.83 ±0.23 | 66.33 ±4.58 | 72.66 ±2.26 | 90.37 ±1.52 |
| HyperGNN | 79.39 ±1.36 | 72.45 ±1.16 | 86.44 ±0.44 | 82.64 ±1.65 | 91.03 ±0.20 | 48.59 ±4.52 | 61.39 ±2.96 | 91.26 ±1.15 |
| SheafHyperGNN | **81.30 ±1.70** | **74.71 ±1.23** | 87.68 ±0.60 | **85.52 ±1.28** | 91.59 ±0.24 | **68.73 ±4.68** | **73.84 ±2.30** | 91.81 ±1.60 |

A general layer of Sheaf Hypergraph Network is defined as:

$$Y = \sigma((I_{nd} - \overset{\bullet}{\Delta})(I_n \otimes W_1)\tilde{X}W_2).$$

Here, $\overset{\bullet}{\Delta}$ can be either $\Delta^{\mathcal{F}} = D^{-\frac{1}{2}}\mathcal{L}^{\mathcal{F}}D^{-\frac{1}{2}}$ for the *linear* sheaf hypergraph Laplacian introduced in Eq. 1 or $\bar{\Delta}^{\mathcal{F}} = D^{-\frac{1}{2}}\bar{\mathcal{L}}^{\mathcal{F}}D^{-\frac{1}{2}}$ for the *non-linear* sheaf hypergraph Laplacian introduced in Eq. 2. Both $W_1 \in \mathbb{R}^{d\times d}$ and $W_2 \in \mathbb{R}^{f\times f}$ are learnable parameters, while $\sigma$ represents ReLU non-linearity.

**Sheaf Hypergraph Neural Network** (SheafHyperGNN). This model utilizes the *linear* sheaf hypergraph Laplacian $\overset{\bullet}{\Delta} = \Delta^{\mathcal{F}}$. When the sheaf is trivial ($d = 1$ and $\mathcal{F}_{v \trianglelefteq e} = 1$), and $W_1 = \mathbf{I}_d$, the SheafHyperGNN is equivalent to the conventional HyperGNN architecture [11]. However, by increasing dimension $d$ and adopting dynamic restriction maps, our proposed SheafHyperGNN becomes more expressive. For every adjacent node-hyperedge pair $(v, e)$, we use a $d \times d$ block matrix to discern each node's contribution instead of a fixed weight that only stores the incidence relationship. The remaining operations are similar to those in HyperGNN [11]. More details on how the block matrices $\mathcal{F}_{v \trianglelefteq e}$ are learned can be found in the following subsection.

**Sheaf Hypergraph Convolutional Network** (SheafHyperGCN). This model employs the non-linear Laplacian $\overset{\bullet}{\Delta} = \bar{\Delta}^{\mathcal{F}}$. Analogous to the linear case, when the sheaf is trivial and $W_1 = \mathbf{I}_d$ we obtain the classical HyperGCN architecture [37]. In our experiments, we will use an approach similar to that in [37] and adjust the Laplacian to include mediators. This implies that we will not only connect the two most discrepant nodes but also create connections between each node in the hyperedge and these two most discrepant nodes, resulting in a denser associated graph. For more information on this variation, please refer to [37] or Supplementary Material.

In summary, the models introduced in this work, SheafHyperGNN and SheafHyperGCN serve as generalisations of the classical HyperGNN [11] and HyperGCN [37]. These new models feature a more expressive implicit regularisation compared to their traditional counterparts.

**Learnable Sheaf Laplacian.** A key advantage of Sheaf Hypergraph Networks lies in attaching and processing a more complex structure (sheaf) instead of the original standard hypergraph. Different sheaf structures can be associated with a single hypergraph, and accurately modeling the most suitable structure is crucial for obtaining effective and meaningful representation. In our proposed models, we achieve this by designing learnable restriction maps. For a d-dimensional sheaf, we predict the restriction maps for each pair of incident (vertex v, hyperedge e) as $\mathcal{F}_{v \trianglelefteq e} = \text{MLP}(x_v || h_e) \in \mathbb{R}^{d^2}$, where $x_v$ represent node features of $v$, and $h_e$ represents features of the hyperedge $e$. This vector representation is then reshaped into a $d \times d$ block matrix representing the linear restriction map for the $(v, e)$ pair. When hyperedge features $h_e$ are not provided, any permutation-invariant operation can be applied to obtain hyperedge features from node-level features. We experiment with three types of $d \times d$ block matrices: diagonal, low-rank and general matrices, with the diagonal version

---

[1]Results where rerun compared to [50] using the same hyperparameters, to fix an existing issue in the original code.

Table 2: **Ablation study on Restriction Maps**: we explore three types of $d \times d$ restriction maps: diagonal, low-rank and general. Diagonal matrices consistently achieve better accuracy on most of the datasets, demonstrating a superior balance between complexity and expressivity

| Name | Cora | Citeseer | Pubmed | Cora_CA | DBLP_CA | Senate | House | Congress |
|---|---|---|---|---|---|---|---|---|
| Diag-SheafHyperGCN | 80.06 ±1.12 | 73.27 ±0.50 | 87.09 ±0.71 | 83.26 ±1.20 | 90.83 ±0.23 | 66.33 ±4.58 | 72.66 ±2.26 | 90.37 ±1.52 |
| LR-SheafHyperGCN | 78.70 ±1.14 | 72.14 ±1.09 | 86.99 ±0.39 | 82.61 ±1.28 | 90.84 ±0.29 | 66.76 ±4.58 | 70.70 ±2.23 | 84.88 ±2.31 |
| Gen-SheafHyperGCN | 79.13 ±0.85 | 72.54 ±2.3 | 86.90 ±0.46 | 82.54 ±2.08 | 90.57 ±0.40 | 65.49 ±5.17 | 71.05 ±2.12 | 82.14 ±2.81 |
| Diag-SheafHyperGNN | 81.30 ±1.70 | 74.71 ±1.23 | 87.68 ±0.60 | 85.52 ±1.28 | 91.59 ±0.24 | 68.73 ±4.68 | 73.62 ±2.29 | 91.81 ±1.60 |
| LR-SheafHyperGNN | 76.65 ±1.41 | 74.05 ±1.34 | 87.09 ±0.25 | 77.05 ±1.00 | 85.13 ±0.29 | 68.45 ±2.46 | 73.84 ±2.30 | 74.83 ±2.32 |
| Gen-SheafHyperGNN | 76.82 ±1.32 | 74.24 ±1.05 | 87.35 ±0.34 | 77.12 ±1.14 | 84.99 ±0.39 | 68.45 ±4.98 | 69.47 ±1.97 | 74.52 ±1.27 |

consistently outperforming the other two. These restriction maps are further used to define the sheaf hypergraph Laplacians (Def. 2, or 4) used in the final Sheaf Hypergraph Networks. Please refer to the Supplementary Material for more details on how we constrain the restriction maps.

# 4 Experimental Analysis

We evaluate our model on eight real-world datasets that vary in domain, scale, and heterophily level and are commonly used for benchmarking hypergraphs. These include Cora, Citeseer, Pubmed, Cora-CA, DBLP-CA [37], House [52], Senate and Congress [53]. To ensure a fair comparison with the baselines, we follow the same training procedures used in [50] by randomly splitting the data into $50\%$ training samples, $25\%$ validation samples and $25\%$ test samples, and running each model 10 times with different random splits. We report average accuracy along with the standard deviation.

Additionally, we conduct experiments on a set of synthetic heterophilic datasets inspired by those introduced by [50]. Following their approach, we generate a hypergraph using the contextual hypergraph stochastic block model [54–56], containing 5000 nodes: half belong to class 0 while the other half to class 1. We then randomly sample 1000 hyperedges with a cardinality 15, each containing exactly $\beta$ nodes from class 0. The heterophily level is computed as $\alpha = \min(\beta, 15 - \beta)$. Node features are sampled from a label-dependent Gaussian distribution with a standard deviation of 1. As the original dataset is not publicly available, we generate our own set of datasets by varying the heterophily level $\alpha \in \{1 \ldots 7\}$ and rerun their experiments for a fair comparison.

The experiments are executed on a single NVIDIA Quadro RTX 8000 with 48GB of GPU memory. Unless otherwise specified, our results represent the best performance obtained by each architecture using hyper-parameter optimisation with random search. Details on all the model choices and hyper-parameters can be found in the Supplementary Material.

**Laplacian vs Sheaf Laplacian.** As demonstrated in the previous section, SheafHyperGNN and SheafHyperGCN are generalisations of the standard HyperGNN [11] and HyperGCN [37], respectively. They transition from the trivial sheaf ($d = 1$ and $\mathcal{F}_{v \triangleleft e} = 1$) to more complex structures ($d \geq 1$ and $\mathcal{F}_{v \triangleleft e}$ a $d \times d$ learnable projection). The results in Table 1 and Table 3 show that both models significantly outperform their counterparts on all tested datasets. Among our models, the one based on linear Laplacian (SheafHyperGNN) consistently outperforms the model based on non-linear Laplacian (SheafHyperGCN) across all datasets. This observation aligns with the performance of the models based on standard hypergraph Laplacian, where HyperGCN is outperformed by HyperGNN in all but two real-world datasets, despite their theoretical advantage [48].

**Comparison to recent methods.** We also compare to several recent models from the literature such as HCHA [38], HNHN [36], AllDeepSets [42], AllSetTransformer [42], UniGCNII [57], HyperND [58], and ED-HNN [50]. Our models achieve competitive results on all real-world datasets, with state-of-the art performance on five of them (Table 1). These results confirm the advantages of using the sheaf Laplacians for processing hypergraphs. We also compare our models against a series baselines on the synthetic heterophilic dataset. The results are shown in Table 3. Our best model, SheafHyperGNN, consistently outperforms the other models across all levels of heterophily. Note that, our framework enhancing classical hypergraph processing with sheaf structure is not restricted to the two traditional models tested in this paper (HyperGNN and HyperGCN). Most of the recent state-of-the-art methods, such as ED-HNN, could be easily adapted to learn and process our novel cellular sheaf hypergraph instead of the standard hypergraph, leading to further advancement in the hypergraph field.

Figure 2: **Impact of Depth and Stalk Dimension** evaluated on the heterophilic dataset ($\alpha = 7$). SheafHyperGNN's performance is unaffected by increasing depth, and high-dimensional stalks is essential for achieving top performance. The Dirichlet energy shows that, while HyperGNN enforces the nodes to be similar, our SheafHyperGNN does not suffer from this limitation, encouraging features diversity.

Table 3: **Accuracy on Synthetic Datasets with Varying Heterophily Levels**: Across all different level of heterophily ($\alpha$), our sheaf-based methods SheafGCN and SheafHGNN consistently outperform their counterparts. Additionally, they achieve top results for all heterophily levels, further demonstrating their effectiveness. For each experiment, the result represents average accuracy over 10 runs.

| Name | heterophily ($\alpha$) | | | | | | |
| --- | --- | --- | --- | --- | --- | --- | --- |
| | 1 | 2 | 3 | 4 | 5 | 6 | 7 |
| HyperGCN | 83.9 | 69.4 | 72.9 | 75.9 | 70.5 | 67.3 | 66.5 |
| HyperGNN | 98.4 | 83.7 | 79.4 | 74.5 | 69.5 | 66.9 | 63.8 |
| HCHA | 98.1 | 81.8 | 78.3 | 75.88 | 74.1 | 71.1 | 70.8 |
| ED-HNN | 99.9 | 91.3 | 88.4 | 84.1 | 80.7 | 78.8 | 76.5 |
| SheafHGCN | **100** | 87.1 | 84.8 | 79.2 | 78.1 | 76.6 | 75.5 |
| SheafHGNN | **100** | **94.2** | **90.8** | **86.5** | **82.1** | **79.8** | **77.3** |

In the following sections, we conduct a series of ablation studies to gain a deeper understanding of our models. We will explore various types of restriction maps, analyze how performance changes when varying the network depth and study the importance of stalk dimension for final accuracy.

**Investigating the Restriction Maps.** Both linear and non-linear sheaf hypergraph Laplacians rely on attaching a sheaf structure to the hypergraph. For a cellular sheaf $\mathcal{F}$ with vertex stalks $\mathcal{F}(v) = \mathbb{R}^d$ and hyperedge stalks $\mathcal{F}(e) = \mathbb{R}^d$ as used in our experiments, this involves inferring the restriction maps $\mathcal{F}_{v \trianglelefteq e} \in \mathbb{R}^{d \times d}$ for each incidence pair $(v, e)$. We implement these as a function dependent on corresponding nodes and hyperedge features: $\mathcal{F}_{v \trianglelefteq e} = \mathrm{MLP}(x_v || h_e) \in \mathbb{R}^{d^2}$. Learning these matrices can be challenging; therefore, we experimented with adding constraints to the type of matrices used as restriction maps. In Table 2 we show the performance obtained by our models when constraining the restriction maps to be either diagonal (Diag-SheafHyperNN), low-rank (LR-SheafHyperNN) or general matrices (Gen-SheafHyperNN). We observe that the sheaves equipped with diagonal restriction maps perform better than the more general variations. We believe that the advantage of the diagonal restriction maps is due to easier optimization, which overcomes the lose in expressivity. More details about predicting constrained $d \times d$ matrices can be found in the Supplementary Material.

**Importance of Stalk Dimension.** The standard hypergraph Laplacian corresponds to a sheaf Laplacian with $d = 1$ and $\mathcal{F}_{v \trianglelefteq e} = 1$. Constraining the stalk dimension to be 1, but allowing the restriction maps to be dynamically predicted, becomes similar to an attention mechanism [38]. However, attention models are restricted to guiding information via a scalar probability, thus facing the same over-smoothing limitations as traditional HyperGNN in the heterophilic setup. Our d-dimensional restriction maps increase the model's expressivity by enabling more complex information transfer between nodes and hyperedges, tailored for each individual pair. We validate this experimentally on the synthetic heterophilic dataset, using the diagonal version of the models, which achieves the best performance in the previous ablation. In Figure 2, we demonstrate how performance significantly improves when allowing higher-dimensional stalks ($d > 1$). These results are consistent for both linear sheaf Laplacian-based models (SheafHyperGNN) and non-linear ones (SheafHyperGCN).

**Influence of Depth.** It is well-known that stacking many layers in a hypergraph network can lead to a decrease in model performance, especially in the heterophilic setup. This phenomenon, called over-smoothing, is well-studied in both graph [59] and hypergraph literature [17]. To analyse the extent to which our model suffers from this limitation, we train a series of models on the most heterophilic version of the synthetic dataset ($\alpha = 7$). For both SheafHyperGNN and its HyperGNN equivalent, we vary the number of layers between $1 - 8$. In Figure 2, we observe that while HyperGNN exhibits a drop in performance when going beyond 3 layers, SheafHyperGNN's performance remains mostly constant. Similar results were observed for the non-linear version when comparing SheafHyperGCN with HyperGCN (results in Supplementary Material). These results indicates potential advantages of our models in the heterophilic setup by allowing the construction of deeper architectures.

**Investigating Features Diversity.** Our theoretical analysis shows that, while conventional Hypergraph Networks tend to produce similar features for neighbouring nodes, our Sheaf Hypergraph

Networks reduce the distance between neighbouring nodes in the more complex hyperedge stalk space. As a result, the nodes' features do not become uniform, preserving their individual identities. We empirically evaluate this, by computing the Dirichlet energy for HyperGNN and SheafHyperGNN (shaded area in Figure 2), as a measure of similarity between neighbouring nodes. The results are aligned with the theoretical analysis: while increasing depth in HyperGNN creates uniform features, SheafHyperGNN does not suffer from this limitation, encouraging diversity between the nodes.

## 5 Conclusion

In this paper we introduce the cellular sheaf for hypergraphs, an expressive tool for modelling higher-order relations build upon the classical hypergraph structure. Furthermore, we propose two models capable of inferring and processing the sheaf hypergraph structure, based on linear and non-linear sheaf hypergraph Laplacian, respectively. We prove that the diffusion processes associated with these models induce a more expressive implicit regularization, extending the energies associated with standard hypergraph diffusion. This novel architecture generalizes classical Hypergraph Networks, and we experimentally show that it outperform existing methods on several datasets. Our technique of replacing the hypergraph Laplacian with a sheaf hypergraph Laplacian in both HyperGNN and HyperGCN establishes a versatile framework that can be employed to "sheafify" other hypergraph architectures. We believe that sheaf hypergraphs can contribute to further advancements in the rapidly evolving hypergraph community, extending far beyond the results presented in this work.

**Acknowledgment**   The authors would like to thank Ferenc Huszár for fruitful discussions and constructive suggestions during the development of the paper and Eirik Fladmark and Laura Brinkholm Justesen for fixing a minor issue in the original HyperGCN code, which led to improved results in the baselines. Iulia Duta is a PhD student funded by a Twitter scholarship. This work was also supported by PNRR MUR projects PE0000013-FAIR, SERICS (PE00000014), Sapienza Project FedSSL, and IR0000013-SoBigData.it.

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
