# Appendix: Sheaf Hypergraph Networks

**Iulia Duta**
University of Cambridge
id366@cam.ac.uk

**Giulia Cassarà**
University of Rome, La Sapienza
giulia.cassara@uniroma1.it

**Fabrizio Silvestri**
University of Rome, La Sapienza
fabrizio.silvestri@uniroma1.it

**Pietro Liò**
University of Cambridge
pl219@cam.ac.uk

In this appendix, we delve into various aspects related to the methodology, including broader impact and potential limitations, proofs for the theoretical results, technical details of the models, and additional experiments mentioned in the main paper. The code associated with the paper will be released soon.

- **Section A** explores the social impact and limitations of our approach, along with a discussion on potential future improvements for the model.
- **Section B** provides proofs for the two primary results presented in the main paper.
- **Section C** offers further information about the method and its training process.
- **Section D** presents additional ablation studies concerning the performance of SheafHyper-GCN as the depth increases. We also provide a comprehensive version of the synthetic heterophilic experiments featured in Table 3 of the main paper, including the standard deviation for all experiments.

## 1 Broader Impact & Limitations

In this paper we introduce a framework that enhances hypergraphs with additional structure, called cellular hypergraph sheaf, and examine the benefits that arise from this approach. Both theoretically and empirically, we demonstrate that this design choice results in a more expressive method for processing higher-order relations compared to its counterpart. Our model is designed as a generic method for higher-order processing without specific components tailored for particular tasks. We test our model on standard benchmark datasets previously used in the literature, as well as synthetically generated datasets created for academic purposes. Consequently, we believe that our paper does not contribute to any specific negative social impacts compared to other hypergraph models.

Our objective is to better incorporate the existing structure in the data. While higher-order connectivity is provided to us in the form of hypergraph structure, we lack access to a ground-truth sheaf associated with the data. Our experiments suggest that jointly learning this structure with the classification task yields top results on several benchmarks. However, without a ground-truth object, it is difficult to determine if the optimisation process results in the ideal sheaf structure. This phenomenon is evident in our experiments comparing different types of restriction maps. Although theoretically less expressive, the diagonal restriction maps generally outperform the low-rank and general ones. This can be attributed to two potential causes: 1) a simpler, diagonal sheaf may be sufficiently complex for our downstream task, or 2) a more easily optimized sheaf predictor needs to be developed to further enhance performance. It is worth noting that similar behavior has been observed in literature for graph-based sheaf models, with the diagonal sheaf consistently achieving good results.

Furthermore, our predictor heavily depends on the quality of the node and hyperedge features. The benchmark datasets we use do not provide hyperedge features, necessitating their inference based on node features. We believe that developing more effective methods for extracting hypergraph

37th Conference on Neural Information Processing Systems (NeurIPS 2023).

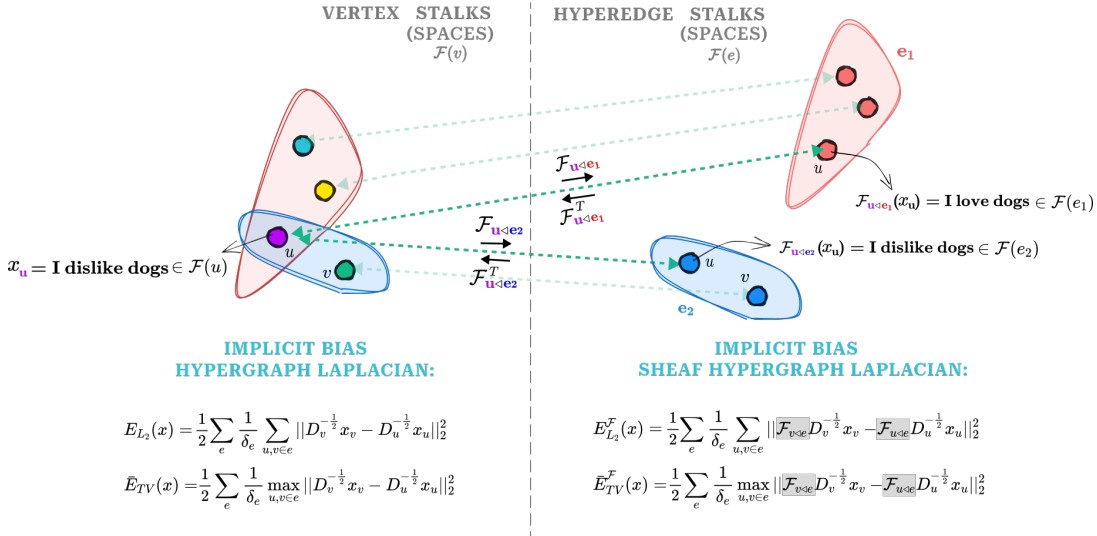

Figure 1: **Visualisation of a hypergraph sheaf and the inductive biases associated with its Laplacian.** For a hypergraph, each node has associated a vertex stalk $F(v) = \mathbb{R}^d$, and each hyperedge has associated a hyperedge stalk $F(e) = \mathbb{R}^d$. For each incident pair $(v, e)$ we can move from the vertex stalk $F(v)$ to the hyperedge stalk $F(e)$ via a linear map $\mathcal{F}_{v \triangleleft e} : \mathbb{R}^d \rightarrow \mathbb{R}^d$. Hypergraph Networks implicitly minimize an energy function defined on the vertex space, aiming to bring together the representation of the neighbouring nodes (**left**). Differently, our Sheaf Hypergraph Networks aim to reduce the discrepancy between the neighbouring representations in the hyperedge space (**right**). This has several advantages. Firstly, we prevent the features from becoming uniform by minimizing the distance in a more complex space, as confirmed both theoretically and empirically. Moreover, in the hyperedge space, each node can have a different representation for each hyperedge it is part of, leading to a more expressive and general message passing framework.

features, along with improved techniques for learning the associated sheaf structure, can unlock the full potential of our model and yield even better results.

Our theoretical framework focuses solely on characterizing the expressivity, demonstrating that SheafHNN can model a broader range of functions than classical HNN. It is worth noting, however, that having a more expressive model does not always ensure better generalizability on test data. Despite this, our empirical results consistently show an advantage of using SheafHNN over HNN across all the tested datasets, which hints that the model can perform well in terms of generalization. Nonetheless, we currently lack any theoretical analyses to substantiate this conclusion.

## 2  Proofs

### 2.1  Proof of Proposition 1

**Proposition 1.** *The diffusion process using a symmetric normalised version of the linear sheaf hypergraph Laplacian minimizes the sheaf Dirichlet energy of a signal $x$ on a hypergraph $\mathcal{H}$. Moreover, the energy decreases with each layer of diffusion.*

*Proof.* We will first demonstrate that a single layer of the diffusion process $Y = (I - \Delta^{\mathcal{F}})X$ using symmetric normalised version of the linear sheaf hypergraph Laplacian $\Delta^{\mathcal{F}} = D^{-\frac{1}{2}} \mathcal{L}^{\mathcal{F}} D^{-\frac{1}{2}} \in \mathbb{R}^{nd \times nd}$ reduces the sheaf Dirichlet energy as follow: $E_{L_2}^{\mathcal{F}}(Y) \leq \lambda_* E_{L_2}^{\mathcal{F}}(X)$, with $\{\lambda_i\}$ the eigenval-

ues of $\Delta^{\mathcal{F}}$ and $\lambda_* = \max_{i;\lambda_i \neq 0} \{(1-\lambda_i)^2\} < 1$. In this manner, as the number of layers approaches infinity, the energy will converge to the minimum energy of 0.

To prove that $E_{L_2}^{\mathcal{F}}(Y) \leq \lambda_* E_{L_2}^{\mathcal{F}}(X)$ we employ the same technique used to demonstrate that the energy associated with the standard hypergraph diffusion decreases with each layer [17]. We will first rewrite the energy as a quadratic term, then decompose it in the eigenvector space and bound it using the eigenvalues upper bounds.

We initiate our proof by establishing a series of Lemmas that will be utilized in our argument.

**Lemma 1.** $E_{L_2}^{\mathcal{F}}(x) = \frac{1}{2}\sum_e \frac{1}{\delta_e} \sum_{u,v \in e} ||\mathcal{F}_{v \trianglelefteq e} D_v^{-\frac{1}{2}} x_v - \mathcal{F}_{u \trianglelefteq e} D_u^{-\frac{1}{2}} x_u||_2^2 = x^T \Delta^{\mathcal{F}} x$

*Proof.*

$$
\begin{aligned}
x^T \Delta^{\mathcal{F}} x &= x^T D^{-\frac{1}{2}} \mathcal{L}^{\mathcal{F}} D^{-\frac{1}{2}} x \\
&= \sum_e \frac{1}{\delta_e} \left( \sum_{v \in e} x_v^T D_v^{-\frac{1}{2}} \mathcal{F}_{v \trianglelefteq e}^T \mathcal{F}_{v \trianglelefteq e} D_v^{-\frac{1}{2}} x_v - \sum_{\substack{w,z \in e \\ w \neq z}} x_w^T D_w^{-\frac{1}{2}} \mathcal{F}_{w \trianglelefteq e}^T \mathcal{F}_{z \trianglelefteq e} D_z^{-\frac{1}{2}} x_z \right) \\
&= \frac{1}{2} \sum_e \frac{1}{\delta_e} \left( \sum_{w \in e} x_w^T D_w^{-\frac{1}{2}} \mathcal{F}_{w \trianglelefteq e}^T \mathcal{F}_{w \trianglelefteq e} D_w^{-\frac{1}{2}} x_w + \sum_{z \in e} x_z^T D_z^{-\frac{1}{2}} \mathcal{F}_{z \trianglelefteq e}^T \mathcal{F}_{z \trianglelefteq e} D_z^{-\frac{1}{2}} x_z \right. \\
&\qquad \left. - \sum_{\substack{w,z \in e \\ w \neq z}} x_w^T D_w^{-\frac{1}{2}} \mathcal{F}_{w \trianglelefteq e}^T \mathcal{F}_{z \trianglelefteq e} D_z^{-\frac{1}{2}} x_z - \sum_{\substack{w,z \in e \\ w \neq z}} x_z^T D_z^{-\frac{1}{2}} \mathcal{F}_{z \trianglelefteq e}^T D_e^{-1} \mathcal{F}_{w \trianglelefteq e} D_w^{-\frac{1}{2}} x_w \right) \\
&= \frac{1}{2} \sum_e \frac{1}{\delta_e} \sum_{w,z \in e} \left( x_w^T D_w^{-\frac{1}{2}} \mathcal{F}_{w \trianglelefteq e}^T - x_z^T D_z^{-\frac{1}{2}} \mathcal{F}_{z \trianglelefteq e}^T \right) \left( \mathcal{F}_{w \trianglelefteq e} D_w^{-\frac{1}{2}} x_w - \mathcal{F}_{z \trianglelefteq e} D_z^{-\frac{1}{2}} x_z \right) \\
&= \frac{1}{2} \sum_e \frac{1}{\delta_e} \sum_{w,z \in e} \left( \mathcal{F}_{w \trianglelefteq e} D_w^{-\frac{1}{2}} x_w - \mathcal{F}_{z \trianglelefteq e} D_z^{-\frac{1}{2}} x_z \right)^T \left( \mathcal{F}_{w \trianglelefteq e} D_w^{-\frac{1}{2}} x_w - \mathcal{F}_{z \trianglelefteq e} D_z^{-\frac{1}{2}} x_z \right) \\
&= \frac{1}{2} \sum_e \frac{1}{\delta_e} \sum_{w,z \in e} ||(\mathcal{F}_{v \trianglelefteq e} D_v^{-\frac{1}{2}} x_v - \mathcal{F}_{u \trianglelefteq e} D_u^{-\frac{1}{2}} x_u)||_2^2
\end{aligned}
$$

$\square$

**Lemma 2.** *The eigenvalues of the symmetric normalised linear sheaf Laplacian $\Delta^{\mathcal{F}}$ are in $[0,1]$.*

*Proof.* Let's denote by $\{\lambda_i\}$ the set of eigenvalues of $\Delta^{\mathcal{F}} \in \mathbb{R}^{nd \times nd}$. We want to prove that $\lambda_i \in [0,1]$. Let's consider $\lambda_1 \geq \lambda_2 \ldots \lambda_{nd}$.

$$\lambda_1 = \max_x \frac{<x, \Delta^{\mathcal{F}} x>}{<x, x>}$$

$$= \max_x \frac{\frac{1}{2}\sum_e \frac{1}{\delta_e}\sum_{u,v\in e}||\mathcal{F}_{v\lhd e}D_v^{-\frac{1}{2}}x_v - \mathcal{F}_{u\lhd e}D_u^{-\frac{1}{2}}x_u||_2^2}{<x,x>}$$

$$= \max_x \frac{\sum_e \frac{1}{\delta_e}(\sum_v ||\mathcal{F}_{v\lhd e}D_v^{-\frac{1}{2}}x_v||_2^2 - \sum_{u,v\in e;u\neq v}<\mathcal{F}_{v\lhd e}D_v^{-\frac{1}{2}}x_v, \mathcal{F}_{u\lhd e}D_u^{-\frac{1}{2}}x_u>)}{<x,x>}$$

$$\leq \max_x \frac{\sum_e \frac{1}{\delta_e}(\sum_v ||\mathcal{F}_{v\lhd e}D_v^{-\frac{1}{2}}x_v||_2^2 + \frac{1}{2}\sum_{u,v\in e;u\neq v}(||\mathcal{F}_{v\lhd e}D_v^{-\frac{1}{2}}x_v||_2^2 + ||\mathcal{F}_{u\lhd e}D_u^{-\frac{1}{2}}x_u||_2^2))}{<x,x>}$$

$$\leq \max_x \frac{\sum_e \frac{1}{\delta_e}(\sum_v ||\mathcal{F}_{v\lhd e}D_v^{-\frac{1}{2}}x_v||_2^2 + \frac{1}{2}2(\delta_e - 1)\sum_v ||\mathcal{F}_{v\lhd e}D_v^{-\frac{1}{2}}x_v||_2^2)}{<x,x>}$$

$$\leq \max_x \frac{\sum_v \sum_e x_v^T D_v^{-\frac{1}{2}}\mathcal{F}_{v\lhd e}^T \mathcal{F}_{v\lhd e}D_v^{-\frac{1}{2}}x_v}{<x,x>}$$

$$\leq \max_x \frac{\sum_v x_v^T D_v^{-\frac{1}{2}}\sum_e(\mathcal{F}_{v\lhd e}^T \mathcal{F}_{v\lhd e})D_v^{-\frac{1}{2}}x_v}{<x,x>}$$

$$\leq \max_x \frac{\sum_v x_v^T x_v}{<x,x>} = 1$$

Since $\lambda_k = \frac{<v_k, \Delta^{\mathcal{F}} v_k>}{<v_k, v_k>} \geq 0$ we conclude that $1 \geq \lambda_1 \geq \lambda_2.. \geq \lambda_{nd} \geq 0$ $\hfill\square$

The Lemma above generalises the result characterising the spectrum of hypergraph Laplacian [**?** ] (Theorem 3.2), extending it for the spectrum of sheaf hypergraph Laplacian.

We will now head to prove the main result: $E_{L_2}^{\mathcal{F}}(Y) \leq \lambda_* E_{L_2}^{\mathcal{F}}(X)$

Let's consider $(\lambda_i, v_i)$ the eigenvalue-eigenvector pairs for $\Delta^{\mathcal{F}}$. We can decomposed the hypergraph signal $x \in \mathbb{R}^{nd\times 1}$ in the eigenvector basis as $x = \sum_i c_i v_i$ for some coefficients $c_i$.

From Lemma 1 we have that $E_{L_2}^{\mathcal{F}}(x) = x^T \Delta^{\mathcal{F}} x$ so we can further decomposed it in the eigenvector basis as follow:

$$E_{L_2}^{\mathcal{F}}(x) = x^T \Delta^{\mathcal{F}} x = x^T \sum_i c_i \lambda_i v_i$$

$$= \sum_i c_i^2 \lambda_i v_i$$

On the other hand we have that $E_{L_2}^{\mathcal{F}}((I - \Delta^{\mathcal{F}})x) = x^T(I - \Delta^{\mathcal{F}})^T \Delta^{\mathcal{F}}(I - \Delta^{\mathcal{F}})x$

$\Delta^{\mathcal{F}}$ has $(\lambda_i, v_i)$ as eigenvalues-eigenvectors pairs, thus $(I - \Delta^{\mathcal{F}})$ will have the same set of eigenvectors $v_i$ with corresponding eigenvalues $(1 - \lambda_i)$. Thus we can decomposed the sheaf Dirichlet energy as:

$$E_{L_2}^{\mathcal{F}}((I - \Delta^{\mathcal{F}})x) = x^T(I - \Delta^{\mathcal{F}})^T \Delta^{\mathcal{F}}(I - \Delta^{\mathcal{F}})x$$

$$= \sum_i c_i^2 \lambda_i (1 - \lambda_i)^2 v_i$$

Let's denote by $\lambda_* = max_{i;\lambda_i \neq 0}\{(1 - \lambda_i)^2\} < 1$. Then:

$$E_{L_2}^{\mathcal{F}}((I - \Delta^{\mathcal{F}})x) = \sum_i c_i^2 \lambda_i (1 - \lambda_i)^2 v_i$$

$$\leq \lambda_* \sum_i c_i^2 \lambda_i v_i = \lambda_* E_{L_2}^{\mathcal{F}}(x)$$

From Lemma 2 we have that $\lambda_i \in [0,1] \Rightarrow \lambda_* < 1$

We conclude that:

$$E^{\mathcal{F}}_{L_2}((I - \Delta^{\mathcal{F}})x) \leq \lambda_* E^{\mathcal{F}}_{L_2}(x) < E^{\mathcal{F}}_{L_2}(x)$$

$\square$

## 2.2 Proof of Proposition 2

**Proposition 2.** *The diffusion process using symmetric normalised version of non-linear sheaf hypergraph Laplacian minimizes the sheaf total-variance of a signal $x$ on hypergraph $\mathcal{H}$.*

*Proof.* Our proof is inspired by the proof in [49], which demonstrates that the non-linear hypergraph diffusion minimizes the total variation.

The summary of our proof is as follow: 1) we show that for a given sheaf $\mathcal{F}$ associated with a hypergraph, the sheaf total variation $\bar{E}_{TV}(x)$ is a convex function, and 2) the sheaf non-linear Laplacian represents a subgradient of the sheaf total variance. Since the subgradient method minimizes convex functions, we conclude that the non-linear sheaf diffusion minimizes the sheaf total variation. We explain the steps in more detail below.

The sheaf total variance is defined as follow: $\bar{E}^{\mathcal{F}}_{TV}(x) = \frac{1}{2}\sum_e \frac{1}{\delta_e} \max_{u,v \in e} ||\mathcal{F}_{v \trianglelefteq e} D_v^{-\frac{1}{2}} x_v - \mathcal{F}_{u \trianglelefteq e} D_u^{-\frac{1}{2}} x_u||_2^2$ where $D_v = \sum\limits_{e; v \in e} \mathcal{F}^T_{v \trianglelefteq e} \mathcal{F}_{v \trianglelefteq e}$.

Given a sheaf structure $\mathcal{F}$ associated with the hypergraph $\mathcal{H}$, let's denote by $g_{\mathcal{F}}(x_u, x_v) = ||\mathcal{F}_{u \trianglelefteq e} D_u^{-\frac{1}{2}} x_u - \mathcal{F}_{v \trianglelefteq e} D_v^{-\frac{1}{2}} x_v||$. From the triangle inequality we have that $g_{\mathcal{F}}(x_u, x_v)$ is convex since:

$$||\mathcal{F}_{v \trianglelefteq e} D_v^{-\frac{1}{2}}(\theta x_v + (1-\theta)y_v) - \mathcal{F}_{u \trianglelefteq e} D_u^{-\frac{1}{2}}(\theta x_u + (1-\theta)y_u)|| \leq \theta ||\mathcal{F}_{v \trianglelefteq e} D_v^{-\frac{1}{2}} x_v - \mathcal{F}_{u \trianglelefteq e} D_u^{-\frac{1}{2}} x_u||$$
$$+ (1-\theta)||\mathcal{F}_{v \trianglelefteq e} D_v^{-\frac{1}{2}} y_v - \mathcal{F}_{u \trianglelefteq e} D_u^{-\frac{1}{2}} y_u||$$

The square of a convex positive function is convex. The maximum of a set of convex functions is convex. Thus: $\max_{w,z}\{g_{\mathcal{F}}(x_w, x_z))^2\} = \max_{w,z} ||\mathcal{F}_{w \trianglelefteq e} D_w^{-\frac{1}{2}} x_w - \mathcal{F}_{z \trianglelefteq e} D_z^{-\frac{1}{2}} x_z||_2^2$ is convex. Following the same approach for all hyperedges we have that $\bar{E}^{\mathcal{F}}_{TV}(x) = \frac{1}{2}\sum_e \frac{1}{\delta_e} \max_{u,v \in e} ||\mathcal{F}_{v \trianglelefteq e} D_v^{-\frac{1}{2}} x_v - \mathcal{F}_{u \trianglelefteq e} D_u^{-\frac{1}{2}} x_u||_2^2$ is also a convex function.

$\bar{E}_{TV}(x)$ is a convex, but non-differentiable function. Therefore, it can be minimized using the subgradient method. In the following, we will demonstrate that the symmetric normalized non-linear sheaf Laplacian operator is a subgradient for $\bar{E}_{TV}(x)$.

It is straightforward to establish that:

$$\bar{E}_{TV}(x) = \frac{1}{2}x^T \bar{\Delta}^{\mathcal{F}} x$$

$$x^T \bar{\Delta}^{\mathcal{F}} x = x^T D^{-\frac{1}{2}} \bar{\mathcal{L}}^{\mathcal{F}} D^{-\frac{1}{2}} x$$

$$= \sum_{e; w \sim_e z} \frac{1}{\delta_e} \Big( x_w^T D_w^{-\frac{1}{2}} \mathcal{F}_{w \trianglelefteq e}^T \mathcal{F}_{w \trianglelefteq e} D_w^{-\frac{1}{2}} x_w - x_w^T D_w^{-\frac{1}{2}} \mathcal{F}_{w \trianglelefteq e}^T \mathcal{F}_{z \trianglelefteq e} D_z^{-\frac{1}{2}} x_z \Big)$$

$$= \frac{1}{2} \sum_{e; w \sim_e z} \frac{1}{\delta_e} \Big( x_w^T D_w^{-\frac{1}{2}} \mathcal{F}_{w \trianglelefteq e}^T \mathcal{F}_{w \trianglelefteq e} D_w^{-\frac{1}{2}} x_w + x_z^T D_z^{-\frac{1}{2}} \mathcal{F}_{z \trianglelefteq e}^T \mathcal{F}_{z \trianglelefteq e} D_z^{-\frac{1}{2}} x_z$$

$$- x_w^T D_w^{-\frac{1}{2}} \mathcal{F}_{w \trianglelefteq e}^T \mathcal{F}_{z \trianglelefteq e} D_z^{-\frac{1}{2}} x_z - x_z^T D_z^{-\frac{1}{2}} \mathcal{F}_{z \trianglelefteq e}^T D_e^{-1} \mathcal{F}_{w \trianglelefteq e} D_w^{-\frac{1}{2}} x_w \Big)$$

$$= \frac{1}{2} \sum_{e; w \sim_e z} \frac{1}{\delta_e} \Big( x_w^T D_w^{-\frac{1}{2}} \mathcal{F}_{w \trianglelefteq e}^T - x_z^T D_z^{-\frac{1}{2}} \mathcal{F}_{z \trianglelefteq e}^T \Big) \Big( \mathcal{F}_{w \trianglelefteq e} D_w^{-\frac{1}{2}} x_w - \mathcal{F}_{z \trianglelefteq e}^T D_z^{-\frac{1}{2}} x_z \Big)$$

$$= \frac{1}{2} \sum_{e; w \sim_e z} \frac{1}{\delta_e} \Big( \mathcal{F}_{w \trianglelefteq e} D_w^{-\frac{1}{2}} x_w - \mathcal{F}_{z \trianglelefteq e} D_z^{-\frac{1}{2}} x_z \Big)^T \Big( \mathcal{F}_{w \trianglelefteq e} D_w^{-\frac{1}{2}} x_w - \mathcal{F}_{z \trianglelefteq e} D_z^{-\frac{1}{2}} x_z \Big)$$

$$= \frac{1}{2} \sum_{e; w \sim_e z} \frac{1}{\delta_e} 2 || \mathcal{F}_{w \trianglelefteq e} D_w^{-\frac{1}{2}} x_w - \mathcal{F}_{z \trianglelefteq e} D_z^{-\frac{1}{2}} x_z ||_2^2$$

$$= \sum_e \frac{1}{\delta_e} \max_{w,z} || \mathcal{F}_{w \trianglelefteq e} D_w^{-\frac{1}{2}} x_w - \mathcal{F}_{z \trianglelefteq e} D_z^{-\frac{1}{2}} x_z ||_2^2$$

$$= 2 \bar{E}_{TV}(x)$$

When the distances $g_{\mathcal{F}}(x_w, x_z)$ are distinct for each pair $w, z \in e$, the function $\bar{E}_{TV}(x)$ is differentiable and $\bar{\Delta}^{\mathcal{F}} x$ represent its gradient (since it is the gradient of the quadratic form $\frac{1}{2} x^T \bar{\Delta}^{\mathcal{F}} x$). So, we only need to find the subgradient for the points where the function is not differentiable (the points where, for a hyperedge, several different pairs of nodes achieve maximum distance).

It is known that when $f$ represent the maximum over a set of convex functions $f_i$, the set of all subgradients of $f$ is determined as $\partial f(x) = \text{conv} \cup_{\alpha \in \mathcal{A}(x)} \partial f_\alpha(x)$ with $\mathcal{A}(x) = \{\alpha | f_\alpha(x) = f(x)\}$. Intuitively, for an hyperedge $e$, if we have more than one pair of nodes achieving maximum distance, computing the derivative for any of these pairs would give us a subgradient of the function. This means that, by following our approach of breaking the ties randomly inside $\bar{\Delta}^{\mathcal{F}} x$ leads to a valid subgradient for those points where the function is non-differentiable.

Given that $\bar{E}_{TV}(x)$ is convex non-differentiable, with the non-linear sheaf diffusion operator described by $\bar{\Delta}^{\mathcal{F}} x = \sum_{e; u \sim_e v} \frac{1}{\delta_e} D_v^{-\frac{1}{2}} \mathcal{F}_{v \trianglelefteq e}^T \big( \mathcal{F}_{v \trianglelefteq e} D_v^{-\frac{1}{2}} x_v - \mathcal{F}_{u \trianglelefteq e} D_u^{-\frac{1}{2}} x_u \big)$ as a subgradient, we conclude that the diffusion process using the non-linear sheaf diffusion operator minimizes the total variance $\bar{E}_{TV}(x)$.

$\square$

## 3 Experimental details

### 3.1 Datasets.

We are running experiments on a set of real-world benchmarking datasets and also on a synthetic set of datasets as in [50]. We provide here details about both of them.

**Real-world datasets.** For the real-world datasets, we test our model on Cora, Citeseer, Pubmed, Cora-CA, and DBLP-CA from [37], as well as House [52], Senate, and Congress [53]. These datasets encompass various network types, including citation networks, co-authorship networks, and political affiliation networks, featuring diverse node and hyperedge features. In co-citation networks (Cora, Citeseer, Pubmed), all documents cited by a specific document are connected through a hyperedge [37]. For co-authorship networks (Cora-CA, DBLP), all documents co-authored by a specific author form a single hyperedge [37]. Node features in both citation and co-authorship networks are represented by the bag-of-words of the associated documents, while node labels correspond to paper classes. In the House dataset, each node represents a member of the US House of Representatives, with hyperedges grouping members belonging to the same committee. For the Congress and Senate

datasets, we adhere to the same settings as described in [50]. In the Congress dataset, nodes symbolize US Congress persons, and hyperedges comprise the sponsor and co-sponsors of legislative bills introduced in both the House of Representatives and the Senate. In the Senate dataset, nodes once again represent US Congress persons, but hyperedges consist of the sponsor and co-sponsors of bills introduced exclusively in the Senate. Each node in both datasets is labeled with the respective political party affiliation.

While reproducing the HyperGCN baseline as reported in [42, 50], we uncovered an essential aspect related to self-loops. The weight coefficient calculation, given by $\frac{1}{2|e|-3}$, yields a value of $-1$ for self-loops when $|e| = 1$. To rectify this, we experimented with an alternative formula: $\max(1, \frac{1}{2|e|-3})$ for determining the weight coefficient, diverging from the original implementation in the [42] repository. By incorporating this modification, we recorded substantial enhancement in HyperGCN's performance, especially within the heterophilic setup. For fairness reasons we decided to report these results as opposed to the original ones in Table 1 of the main paper.

**Synthetic datasets.** We generate the synthetic heterophilic dataset inspired by the ones introduced by [50]. We generate a 5000 nodes hypergraph using the contextual hypergraph stochastic block model [54, 55, 56]. Half of these nodes belong to class 0 while the other half to class 1, and task is formulated as a node-classification problem. We randomly sample 1000 hyperedges, each with a cardinality of 15, to create the hyperedges. Each hyperedge contains exactly $\beta$ nodes from class 0 and $15 - \beta$ nodes from class 1. The heterophily level is computed as $\alpha = \min(\beta, 15 - \beta)$. To create node features, we sample from a label-dependent Gaussian distribution with a standard deviation of 1. Since the original dataset is not publicly available, we generate our own set of 7 datasets, by varying the heterophilic level $\alpha \in \{1 \dots 7\}$.

### 3.2 Implementation details.

We define a layer of Sheaf Hypergraph Network as used in both the linear (SheafHyperGNN) and non-linear (SheafHyperGCN) version of the architecture as follow:

$$Y = \sigma((I_{nd} - \overset{\bullet}{\Delta})(I_n \otimes W_1)\tilde{X}W_2) \tag{1}$$

For a hypergraph with $n$ nodes, we denote by $\tilde{X} \in \mathbb{R}^{nd \times f}$ the representations of the nodes in the vertex stalks of a $d$-dimensional sheaf. We note that instead of each node being represented as a row in the feature matrix, as is common in standard HNNs, in our SheafHNN each node is characterized by $d$ rows, resulting in a $d \times f$ feature matrix for each node.

In Equation. 1 the features corresponding to each node are processed in a factorized way: $W_1 \in \mathbb{R}^{d \times d}$ is used to combine the $d$ dimensions of the vertex stalk, independently for each node and each channel, while $W_2 \in \mathbb{R}^{f \times f}$ corresponds to the usual linear projection combining the $f$ features, independently for each node and each stalk dimension. $\sigma$ denotes ReLU non-linearity.

In our codebase, for fair comparison against other baselines, we use the same pipeline from the public repository of [42] and [50]. We also use some auxiliary functions from [22]. All models are trained in PyTorch [?], using Adam optimizer [?] for 100 epochs. For all sheaf-based experiments, we observed a much faster convergence than the usual HNNs.

The results reported in Table 1 and Table 2 of the main paper are obtained using random hyper-parameter tuning. We search the following sets of hyper-parameters:

1. *Stack dimension $d$* from $1 - 8$

2. The sheaf structure is either *shared between layers or recompute* every layer based on the intermediate representations. We noticed that the fixed one is, in general, easier to optimize.

3. $W_1$ is either a *learnable parameter or fixed* to the identity matrix.

4. The *type of normalisation* used for the Laplacian is either symmetric $\Delta = D^{-\frac{1}{2}}\mathcal{L}D^{-\frac{1}{2}}$ or asymmetric $\Delta = D^{-1}\mathcal{L}$; based on the degree ($D$ is the degree matrix) or sheaf-based ($D = \sum_{e; v \in e} \mathcal{F}_{v \unlhd e}^T \mathcal{F}_{v \unlhd e}$). In our experiments the degree-based normalisation is, in general, more stable to optimize.

5. Since none of datasets we are using has *hyperedge features*, we are treating the way to generate edge features $(x_e)$ as a hyperparameter. We experiment with the following approaches: a) $x_e = \bigoplus_{v \in e} x_v$; b) $x_e = \bigoplus_{v \in e} h_v$, with $h_v$ the hidden representation of the nodes c) $x_e = \bigoplus_{v \in e} \text{MLP}(x_v)$ or d) more general as in [**?** ] $x_e = \sigma'\left(M\sigma\left(W^T \begin{bmatrix} x_{v_1} \\ 1 \end{bmatrix} \odot \cdots \odot \begin{bmatrix} x_{v_k} \\ 1 \end{bmatrix}\right)\right)$, with $\sigma' = \text{ReLU}$ and $\sigma = \tanh$. Generally, all of them behaves on par.

6. *Non-linearity* for $\mathcal{F}_{v \trianglelefteq e}$ is either sigmoid or tanh.

7. *Learning rate* from $\{0.1, 0.01, 0.001\}$; *weight decay* from $\{0, 1e - 05\}$; *dropout rate* from $\{0.1, 0.2 \ldots 0.9\}$

8. *Number of layers* from $1 - 8$; *hidden dimension* from $\{16, 32, 64, 128, 256, 512\}$

### 3.3 Restriction maps details.

As mentioned in the main paper, the general setup for predicting restriction maps is as follow: for each incidence pair $(v, e)$, we predict a $d \times d$ block matrix $\mathcal{F}_{v \trianglelefteq e} = \text{MLP}(x_v || h_e) \in \mathbb{R}^{d^2}$. We experiment with three types of $d \times d$ restriction maps: diagonal, low-rank and general.

**General restriction maps.**  In the general case, the MLP predicts $d^2$ parameters that we rearrange in a square $d \times d$ matrix representing the restriction maps.

**Diagonal restriction maps.**  For the diagonal restriction maps, the MLP used to predict the restriction maps outputs only $d$ elements. The final restriction block is obtained by creating a matrix with these $d$ elements on the diagonal. While being less expressive than the general case, the diagonal matrix is more efficient both in terms of parameters and computation.

**Low-Rank restriction maps.**  In the low-rank case, to predict a rank-$r$ matrix, $2 * d * r + d$ elements are predicted. They are rearranged in two matrices $A \in \mathbb{R}^{d \times r}$, $B \in \mathbb{R}^{r \times d}$ and a vector $c \in \mathbb{R}^{d \times 1}$. The final restriction matrix is obtained as $AB^T + diag(c)$. When $r < (d - 1)/2$, the low-rank represent a more efficient option in terms of parameters.

In experiments, we observed that models using the diagonal version of the restriction map consistently outperform those using either low-rank or general versions. We believe that finding better, more efficient ways of predicting and optimizing the restriction maps might further improve the results presented in this paper.

### 3.4 Non-linear sheaf Laplacian with mediators.

The non-linear hypergraph Laplacian relies on a much sparser representation compared to the linear hypergraph Laplacian. While the linear one creates $\frac{|e|(|e|-1)}{2}$ edges for each hyperedge, the non-linear one only draws a single connection for each hyperedge. To allow each node to participate in the hypergraph diffusion, while maintaining a sparse representation, [**?** ] introduces a variation of the non-linear Laplacian that propagates flow through all the nodes in a hyperedge and still preserves the theoretical properties of the original non-linear Laplacian.

**Definition 1.** Based on their formulation, we define the *non-linear sheaf hypergraph Laplacian with mediators* as follow:

1. For each hyperedge $e$, compute $(u_e, v_e) = argmax_{u,v \in e}||\mathcal{F}_{u \triangleleft e} x_u - \mathcal{F}_{v \triangleleft e} x_v||$, the set of pairs containing the nodes with the most discrepant features in the hyperedge stalk [1] and the set of mediators $K_e = \{k \in e : k \neq u_e, k \neq v_e\}$

2. Build an undirected graph $\mathcal{G}_H$ containing the same sets of nodes as $\mathcal{H}$ and, for each hyperedge $e$ connects the most discrepant nodes $(u, v)$, and also add a connection between $\{u, v\}$ and all the nodes in $K_e$. (from now on we will write $u \sim_e v$ if they are connected in the $\mathcal{G}_H$ graph due to the hyperedge e). If multiple pairs have the same maximum discrepancy, we will randomly choose one of them.

---

[1]Note that for the normalised version of the Laplacian $x_u \rightarrow D_u^{-\frac{1}{2}} x_u$

Table 1: **Average accuracy and standard deviation on Synthetic Datasets with Varying Heterophily Levels**: Across all different level of heterophily, the sheaf-based methods consistently outperform their counterparts. Additionally, they achieve top results for all heterophily levels, further demonstrating their effectiveness.

| Name | heterophily ($\alpha$) | | | | | | |
|---|---|---|---|---|---|---|---|
| | 1 | 2 | 3 | 4 | 5 | 6 | 7 |
| HyperGCN | 83.9 ±18.6 | 69.4 ±13.0 | 72.9 ±10.7 | 75.9 ±1.0 | 70.5 ±9.7 | 67.3 ±12.1 | 66.5 ±10.8 |
| HyperGNN | 98.4 ±0.3 | 83.7 ±0.6 | 79.4 ±0.9 | 74.5 ±0.8 | 69.5 ±0.9 | 66.9 ±1.13 | 63.8 ±1.1 |
| HCHA | 98.1 ±0.6 | 81.8 ±1.3 | 78.3 ±1.4 | 75.9 ±1.3 | 74.1 ±1.6 | 71.1 ±1.4 | 70.8 ±1.0 |
| ED-HNN | 99.9 ±0.1 | 91.3 ±3.0 | 88.4 ±1.5 | 84.1 ±2.8 | 80.7 ±1.2 | 78.8 ±0.9 | 76.5 ±1.4 |
| SheafHGCN | **100** ±**0.0** | 87.1 ±2.4 | 84.8 ±1.1 | 79.2 ±2.0 | 78.1 ±0.6 | 76.6 ±1.1 | 75.5 ±1.4 |
| SheafHGNN | **100** ±**0.0** | **94.2** ±**0.9** | **90.8** ±**1.1** | **86.5** ±**1.0** | **82.1** ±**1.2** | **79.8** ±**0.7** | **77.3** ±**1.3** |

3. Define the sheaf non-linear hypergraph Laplacian as:

$$\bar{\mathcal{L}}_{\mathcal{F}}(x)_v = \sum_{e;u \sim_e v} \frac{1}{\delta_e} \mathcal{F}_{v \trianglelefteq e}^T \big( \mathcal{F}_{v \trianglelefteq e} x_v - \mathcal{F}_{u \trianglelefteq e} x_u \big) \tag{2}$$

We note that in the non-linear Laplacian with mediators, all the nodes in a hyperedge have an active role in the diffusion. This approach requires a linear number of edges, remaining more efficient than the linear version of the Laplacian.

We use this approach in all the experiments involving non-linear sheaf hypergraph Laplacian.

## 4 Additional experiments

**Full results for the comparison with baselines.** Due to the space contraints in the main paper (Table 3), we only report the average accuracy for the experiments performed on the Synthetic heterophilic datasets (with $\alpha = 7$). In Table 1 of the Appendix, we include both the average performance and the standard deviation obtained across 10 random splits. Both our models, SheafHyperGNN and SheafHyperGCN, outperform their counterparts, with SheafHyperGNN achieving the best results among all the baselines. We note that on the synthetic dataset, HyperGCN is very unstable during training, with a high standard deviation between the runs. We believe this is due to the noisy features that negatively and irreversibly affect the creation of the non-linear Laplacian. In contrast, our generalization achieves more stable results.

**Influence of Depth.** We present the results for varying the number of layers for both SheafHyperGNN and SheafHyperGCN. The observations remain consistent for both the models using linear and non-linear sheaf Laplacian: the performance of traditional HyperGNN and HyperGCN generally decreases when increasing the depth of the model, a phenomenon known in the literature as over-smoothing. On the other hand, the results show that our generalized versions of the model, SheafHyperGNN and SheafHyperGCN respectively, do not suffer from this limitation, allowing for deeper architectures without sacrificing performance.