# OpenReview forum: "Sheaf Hypergraph Networks"
_NeurIPS.cc/2023/Conference — NeurIPS 2023 poster_

### Official Review · Reviewer_XTYg · 2023-06-10

**Soundness:** 3 good
**Presentation:** 3 good
**Contribution:** 3 good
**Rating:** 7
**Confidence:** 4

**Summary:**

\
The primary research focus of this paper is to investigate the potential of neural networks for processing hypergraph datasets.

The paper utilises the mathematical concept of sheaf, which describes how locally defined data on a space can be consistently pieced together, to enhance hypergraphs by associating vector spaces with nodes and hyperedges, facilitating the transfer of information through linear projections.

The paper lays the groundwork for "sheaf" hypergraph Laplacians, examining their diffusion processes to understand the inductive biases they generate, which play a pivotal role in driving the progress of hypergraph neural networks.

**Strengths:**

\
**Clarity**

The introduction of the paper offers a concise and informative overview of the research topic, providing necessary context by proposing the generalisation of hypergraph Laplacians to advanced sheaf Laplacians for capturing intricate phenomena.

It effectively outlines the objectives of advancing neural networks on hypergraphs and sets the stage for the study and development of sheaf hypergraph neural networks.

The submitted work showcases a well-structured and organised presentation, with a logical progression of ideas, clearly defined sections and headings, and enhanced clarity through the inclusion of code and supplementary materials.

\
**Quality**

The paper's claims are supported by sound theoretical analyses of the diffusion processes (e.g. propositions 1, 2) and empirical studies including involving different restriction maps (e.g. Table 2).

By investigating the impact of depth and heterophily levels, the paper reveals additional insights, contributing to its overall quality.

In summary, the methodology, experimental design, and theoretical framework employed to support the contributions are sound, credible, and well-founded.

\
**Originality**

By introducing sheaf Laplacians, the paper tackles a limitation present in current hypergraph Laplacians (e.g. see lines 163-172), which have been instrumental in the development of existing hypergraph networks.

By adopting unique perspectives (e.g. opinion dynamics perspective see lines 173-180), the paper illuminates the nuanced and intricate phenomena that can be captured by the sheaf structure.

Overall, the paper sheds light on a previously unexplored limitation in the existing literature, showcasing its originality.

**Weaknesses:**

\
**Significance**

Recent research has explored sheaf-based approaches for graph data, resulting in the development of advanced graph neural network architectures [1, 2] that draw inspiration from sheaf Laplacians.

Hyperedges in hypergraph datasets can be represented as pairwise connections, treating them as graph datasets, and the linear sheaf Hypergraph Laplacian employed by SheafHyperGNN yields the best results among the models in Table 1.

To strengthen the significance and necessity of a Sheaf operator on higher-order data, it is crucial to conduct comprehensive empirical analyses, such as comparing with the best SheafGNN architecture as a baseline and exploring hypergraph datasets where the non-linear Laplacian captures substantially more information than its linear counterpart.

1. [Sheaf Neural Networks, In TDA and Beyond, 2020](https://openreview.net/forum?id=GgcgIJsT8HD)
2. [Neural Sheaf Diffusion: A Topological Perspective on Heterophily and Oversmoothing in GNNs, In NeurIPS'22](https://openreview.net/forum?id=vbPsD-BhOZ)





**Questions:**

* Does the paper include any experiments demonstrating that a neural network with a non-linear Sheaf Laplacian captures significantly more information than a linear operator? Please clarify if such analysis is already present, and if not, it would be valuable to include as an important contribution to the paper.
* Can we rely solely on the increased expressiveness of sheaf Laplacians, or should we also consider the importance of good generalisability for achieving strong empirical performance on test data?
* In terms of generalisation, while both the AllSet framework [3] and Sheaf Hypergraph Networks generalise and offer advancements over earlier models, does the Sheaf Networks framework have the capacity to encompass all instances of the AllSet framework, or are there any specific scenarios within the AllSet framework that cannot be accommodated by Sheaf Networks?

3. [You are AllSet: A Multiset Function Framework for Hypergraph Neural Networks, In ICLR'22](https://openreview.net/forum?id=hpBTIv2uy_E)


**Limitations:**

Adequately addressed (e.g. see Appendix A).

---

> ### Author Rebuttal · Authors · 2023-08-09
>
> We thank the reviewer for the very detailed review and constructive feedback. We are delighted the reviewers agree that we propose a sound method, with a pivotal role for the hypergraph neural networks progress.
>
> In the following, we discuss the main remarks raised by the reviewer, and we will incorporate all the feedback into the final version of the paper.
>
> **Expressivity vs generalisability**
>
> We entirely agree with the reviewer’s observation that having a more expressive model does not necessarily offer better generalisability on test data and that both expressivity and generalisability are important for good performance. Our theoretical framework focuses solely on characterising the expressivity, demonstrating that SheafHNN are able to model a broader range of functions compared to classical HNN. However, that does not provide any guarantees that the optimization process will identify the optimal sheaf structure for the test data.
>
> The empirical results consistently show an advantage of using SheafHNN compared to HNN across all tested datasets, hinting that the model is able to perform well in terms of generalisability as well. However, we do not have any theoretical analyses in this regard. We will clearly emphasise this in the limitation section. Investigating aspects like generalizability and extrapolation power of SheafHNN represents an extremely promising avenue for future research.
>
> **Importance of non-linear Sheaf Laplacian**
>
> From a practical standpoint, the non-linear Sheaf Laplacian offers the advantage of relying on a sparser connectivity compared to its linear counterpart. In each hypergraph, the non-linear Sheaf Laplacian propagates information along a reduced number of edges, approximately $O(\sum_e 2|e|)$, whereas the linear Sheaf Laplacian requires a much higher number of connections, approximately  $O(\sum_e \frac{(|e|-1)|e|}{2})$., where $|e|$ is the cardinality of hyperedge $e$.
>
> From a theoretical perspective, [1] indicates that the spectral properties of the non-linear Laplacian are more suited for higher-order processing compared to the linear Laplacian. For instance, the non-linear Laplacian leads to a more balanced partition in the minimum cut problem compared to its linear version (min-cut problem is known to be tightly related to the semi-supervised node classification task).  This balanced minimum cut avoids making a cut inside a hyperedge that would result in isolating nodes.  Despite being recognized as a powerful tool for hypergraphs, we were not able to empirically observe these advantages in our experiments. We believe that this could be attributed to the scarcity of datasets specifically dedicated to higher-order analysis.
>
> In our experiments, we observed an advantage of the non-linear SheafHyperGCN over the linear SheafHyperGNN on a subset of datasets, only in the more challenging optimization scenarios, when the predicted sheaf is either *general* or restricted to be *low-rank* (as shown in Table 2). However, we believe that this advantage does not stem from the fact that non-linear models capture significantly more information. Instead, it is more likely due to the easier optimization process facilitated by the sparser computational graph, as mentioned in the first paragraph.
>
>
> **Theoretical comparison against AllSet framework**
>
> In our research, our primary focus was on integrating the sheaf into HGNN and HGCN architectures, enabling us to explore the inductive bias theoretically. This inductive bias is an interpretable element absent in the AllSet framework. The experiments  showing that SheafHGNN not only obtains superior performance compared to HGNN, but also surpasses the more powerful AllSet architectures (AllDeepSets and AllSetTransformers) suggests that the imposed inductive bias helps our model to more effectively learn the downstream task.
>
> The linear SheafHNN utilises the aggregated message function in the form of $f_{e} = \sum_{v \in e} \tilde{h}(x_v, z_e) \times  x_v$, where $\tilde{h}(x_v, z_e)$ represent the sheaf associated with the pair $(v,e)$. Comparing the capacity of SheafHNN against the AllSet framework leads to investigating the universality of this message-passing function, which is an interesting, but highly challenging task. Our theoretical framework does not provide a comparison of this sort. In the current form, there might exist functions that the general AllDeepSet architecture (with an aggregated message in the form $MLP_1(\sum_{v \in e} MLP_2(x_v))$) can represent, while SheafHNN does not.
>
> While being outside the scope of the current work,  we mention that integrating the sheaf component into AllDeepSet is an easy step forward. This integration involves replacing the propagation function $MLP_1(\sum_{v \in e} MLP_2(x_v))$ with $\mathbf{MLP_1}(\sum_{v \in e} h(x_v, z_e) \times \mathbf{MLP_2}(x_v))$, where $h(x_v, z_e) \in \mathbb{R}^{d \times d}$ is the learned sheaf associated with the incident pair $(v,e)$, $x_v \in \mathbb{R}^{d \times f}$ and $\mathbf{MLP_*}$ is applied row-wise. This would guarantee to preserve the theoretical universality of AllSet message passing (since the model will be equivalent to the classical AllSet for the trivial sheaf). However, it comes at the cost of losing the interpretable inductive bias that SheafHNN possesses.
>
> *[1] Hein, Matthias, Setzer, Simon, Jost, Leonardo and Rangapuram, Syama Sundar. The Total Variation on Hypergraphs - Learning on Hypergraphs Revisited.*

---

> > ### Comment · Reviewer_XTYg · 2023-08-15
> > **Rating Retained**
> >
> > Thanks for the work done in handling the concerns. I've decided to keep my rating.

---

### Official Review · Reviewer_2wJ1 · 2023-06-29

**Soundness:** 4 excellent
**Presentation:** 3 good
**Contribution:** 4 excellent
**Rating:** 7
**Confidence:** 4

**Summary:**

This paper generalizes hyper-graph Laplacians to hyper-graphs attached with sheaves, called Hypergraph Sheaf Laplacian, and constructued hyper-graph NNs with proposed Hypergraph Sheaf Laplacian. Experiments demonstrate the effectiveness of SheafHyperGNNs over normal HyperGNNs.

**Strengths:**

I find this paper well-written and easy to follow. The paper addresses two crucial challenges in the GNN community: hyper-graph processing and heterophily graph processing. The concept of hyper-graph sheaf Laplacian looks convincing and promising. The SheafHyperGNN is derived from hyper-graph sheaf Laplacian, which provides a rigorous framework to study the properties of it (e.g. to prove it doesn't suffer from over-smoothing). I believe both the new concept of hyper-graph sheaf Laplacian and the proposed SheafHyperGNN can be benificial to the community.

-----

After reading authors' rebuttal, I decide to still keep my current rating.

**Weaknesses:**

See Questions.

**Questions:**

1. I feel the proposed methods are kind of similar to heterogeneous graph NNs, for example RGCN[1]. I wonder if the authors can discuss the relationship between their proposed methods and those heterogeneous graph NNs, for example can RGCN be viewed as a special case of SheafHyperGNN?

2. Given the SheafHyperGNN has learnable sheaves, will there be an overfitting risk and do the authors have any ideas on how to avoid it? For example I personally have also tried to somehow attatch parameters to each edge in message passing, and I found the GNN simply learns zero message passing weights and degenrates to an MLP and uses the MLP to overfit the training set. Theoretically, it seems the proposed SheafHyperGNN has the same risk and I wonder if the authors observed similar phenomenons in experiments? (say do you need to tune the hyper-parameters to avoid it?) If so it would be nice to discuss potential ways to address it.


[1] Modeling Relational Data with Graph Convolutional Networks

---

> ### Author Rebuttal · Authors · 2023-08-09
>
> We are grateful that the reviewer appreciates our work and believes that the novel direction is promising and beneficial to the community. We thank the reviewer for the constructive feedback and for pointing out to us a relevant related domain.
>
> In the following, we discuss in more detail each point raised by the reviewer. We will incorporate all the feedback and related works into the final version of the paper.
>
> **Connection with the Heterogeneous graph NN literature**
>
> Both SheafHyperNN and heterogeneous methods for graphs such as RGCN [1] share a technical similarity: they use distinct “message functions” for each edge (each type of edge for RGCN). However, they differ in their approach. In contrast to RGCN, SheafHyperNN does not learn different parameters for each incident relationship. Instead, it dynamically predicts a projection (the reduction map) based on (node, hyperedge) features for each relationship. More specifically, the sheaf predictor takes the node and hyperedge features for each incident pair and predicts a $d \times d$ matrix to be used as a linear projection.  The parameters that are learned in our approach are not directly the one used in projection (as in RGCN) but the parameters of the small network used to predict the projection. As a result, the number of parameters for our model does not scale with the number of hyperedges (types of edges in the RGCN scenario). Another significant distinction lies in the types of data each method addresses. SheafHNN is designed for hypergraph representation learning, whereas RGCN is applied to heterogeneous graph data.
>
> We thank the reviewers for pointing out this line of work. We believe that, while the methods are different, there are overlapping motivations driving both directions, making SheafHNN a candidate tool for representing heterogeneous hypergraph data. Exploring this potential through experiments represents an exciting avenue for future research.
>
> **Overfitting risk due to learnable sheaves**
>
> Introducing the learnable sheaf structure in the model leads to an increase in the number of parameters compared to the backbone.  However, we mitigate this by employing a small network to predict the projection, which is shared across all incidence relations. This approach enables us to keep the number of additional parameters relatively small, and independent of the number of relations.
>
> To further address the risk of overfitting, we experimented with various restrictions applied to the projection matrix, such as diagonal, low-rank and orthogonal constraints. These restrictions allow us to predict a smaller number of parameters, thus requiring a smaller predictor network. In line with the reviewer's comment, the empirical advantages of using a diagonal projection over more general forms can be attributed, in part, to a reduced risk of overfitting. However, overfitting can still be a problem especially in very small datasets. We believe that exploring non-parametric ways of predicting the sheaf represents a highly interesting research direction.
>
> *[1] Michael Sejr Schlichtkrull, Thomas N. Kipf, Peter Bloem, Rianne van den Berg, Ivan Titov, Max Welling. Modeling Relational Data with Graph Convolutional Networks.*

---

> ### Comment · Reviewer_2wJ1 · 2023-08-14
>
> Thanks for the authors' response. The reply effectively addressed my questions. Therefore, I have decided to keep my rating as accept.

---

### Official Review · Reviewer_qurz · 2023-07-06

**Soundness:** 2 fair
**Presentation:** 3 good
**Contribution:** 2 fair
**Rating:** 4
**Confidence:** 4

**Summary:**

The paper proposes cellular sheaf to enhance the high-order relationships among hypergraphs. Specifically, it first designs two formulations, linear sheaf hypergraph Laplacian and non-linear sheaf hypergraph Laplacian to construct the hypergraph structure. Then the two sheaf hypergraph Laplacians are leveraged to design two types of models, i.e., sheaf hypergraph neural network and sheaf hypergraph convolutional networks. Detailed theoretical analysis and extensive experiments are provided to demonstrate the effectiveness of the designed model.

**Strengths:**

1. The topic of enhancing the hypergraph structure is impressive.

2. The paper is well-organized overall and easy to follow.

3. Figures are concise, enabling readers to easily catch the key points.

**Weaknesses:**

1. The motivation for the sheaf hypergraph network is not clearly discussed in this work. In the section Introduction, this work does not clearly discuss the existing limitation of extracting the high-order relationships among hypergraphs. Instead, this work directly introduces sheaf for hypergraphs, which is confusing and not consistent from my point of view.


2. The format of the equations and some notations are very confusing in this manuscript. There should have a comma or a full stop after the equations.
For instance, a full stop should be added at the end of equation 2. All equations in Definitions should also be numbered.

3. Some statements in this manuscript are very confusing and vague. There are also a lot of typos and grammar mistakes. For instance, line 31, 'that allows for'--> 'allow for'; In the caption of Figure 1, 'Than' --> 'Then'.

4. In Table 1, SheafHyperGNN gains excellent performance over most benchmark datasets, while the performance of SheafHyperGNN over Congress is not that satisfactory than ED-HNN and the performance gap (i.e., 3\%) between these two models is large. I would like to know the reason behind that.



**Questions:**

1. Can you clearly explain the motivation for this work?
2. Please discuss the existing challenges of hypergraph models in extracting the high-order relationships among hypergraphs.
3. I would like to know the reason behind the performance gap between SheafHyperGNN and ED-HNN over Congress.
4. Grammar mistakes and vague statements should be revised.

**Limitations:**

This work clearly discusses the limitation of the model design.

---

> ### Author Rebuttal · Authors · 2023-08-09
>
> Thank you for your thorough review and constructive feedback on our paper. We appreciate the time you've taken to review our work, and we would like to address your concerns and questions.
>
> **Motivation and challenges**
>
> The motivation for our work comes from recognizing that current hypergraph models struggle to capture higher-order relationships effectively. As described in [1], conventional hypergraph neural networks (HGNNs) often suffer from the  problem of over-smoothing [2]: as we propagate the information inside the hypergraph the representations of the nodes become overly uniform across neighbourhoods. This over-smoothing effect hampers the capability of HGNNs to capture local, higher-order nuances, especially on heterophilic node-classification tasks, when connected nodes tend to belong to different classes.
>
> Our proposed Sheaf Hypergraph Networks aim to overcome these limitations by introducing cellular sheaves into hypergraphs, providing a more expressive and nuanced representation of the underlying data structure. We achieve this by allowing each node to have a different representation for each hyperedge it is part of ($F_{v \trianglelefteq e} x_v$). Moreover, as we shown in Proposition 1 and  2 and empirically in Figure 2, our sheaf diffusion brings closer the representation of the nodes in this hyperedge space as opposed to bringing them closer in the feature space (which is the underlying cause of over-smoothing).
>
> Moreover, conventional hypergraphs lack the additional structure to capture local, higher-order connectivity nuancedly, hence the introduction of cellular sheaves into hypergraphs which enable a more complex flow of messages compared to usual Laplacians.
>
> **Formatting and grammar**
>
> We apologize for any confusion caused by the formatting of equations and notations. We appreciate your suggestion and will ensure all equations are appropriately numbered and punctuated in the revised manuscript.
>
> We value your feedback on the quality of language and will undertake a thorough revision to correct any grammatical errors and confusing statements. We will particularly address the instances you've mentioned in line 31 and the caption of Figure 1.
>
> **Performance**
>
> We believe that the performance gap between SheafHyperGNN and ED-HNN over Congress may be due to the unique structure or properties of the Congress dataset. The unique design of the Congress dataset, which contains more hyperedges and fewer nodes than other datasets, presents a specific challenge for our model. When hyperedge features are not provided, we can typically apply any permutation-invariant operation to obtain hyperedge features from node-level elements. Without a ground-truth sheaf structure associated with the data, SheafHNN heavily relies on the quality of nodes and hyperedge features to infer this structure. However, the Congress dataset does not contain either node or hyperedge attributes. We followed the method of [3] to generate node features from label-dependent Gaussian distribution, as proposed by [4] . Given that the node and hyperedge features are caused predominantly by Gaussian noise, we might end up with a noisier sheaf structure which damages the learning process. We will delve into greater detail about this in the revised manuscript. Designing better features for these highly challenging cases represent an interesting area for future study.
>
> Once again, thank you for your valuable feedback. By addressing these points, we can significantly improve the clarity and quality of the paper.
>
> *[1] Guanzi Chen, Jiying Zhang, Xi Xiao, Yang Li. Preventing Over-Smoothing for Hypergraph Neural Networks.*
>
> *[2] Qimai Li, Zhichao Han, & Xiao-Ming Wu. Deeper Insights Into Graph Convolutional Networks for Semi-Supervised Learning.*
>
> *[3] Eli Chien, Chao Pan, Jianhao Peng, and Olgica Milenkovic. You are allset: A multiset function framework for hypergraph neural networks.*
>
> *[4] Yash Deshpande, Subhabrata Sen, Andrea Montanari, Elchanan Mossel. Contextual stochastic block models.*

---

> > ### Comment · Reviewer_qurz · 2023-08-20
> > **Keep my scores**
> >
> > Thank you for your clarification. But I do not think your clarification has adequately addressed all my concerns and I would like to keep my score.

---

### Official Review · Reviewer_C7Ao · 2023-07-07

**Soundness:** 2 fair
**Presentation:** 3 good
**Contribution:** 2 fair
**Rating:** 4
**Confidence:** 3

**Summary:**

The paper introduces the cellular sheaves to hypergraph and proposes two sheaf hypergraph laplacians. In addition, they also propose two models: sheaf HNN and Sheaf HCN. The experiments demonstrated the effectiveness of the proposed models.

**Strengths:**

- The paper is well-written and well-organized.
- The authors incorporate the sheaves to hypergraph laplacian, which would promote the development of hypergraph spectral theory.
- The proposed two hypergraph neural networks provide new options for hypergraph learning.

**Weaknesses:**

1. The novelty of this paper is limited. It uses the techniques in graph-based methods[1] to existing hypergraph laplacian[2][3]. The Sheaf hypergraph networks also have the same form of Sheaf Convolutional Network (SCN). The Sheaf Dirichlet energy and its convergence property also can be trivially generalized from [4].
2. The experiments don’t bring surprising results. The replacement of trivial hypergraph Laplacian with sheaf laplacian can improve the performance, which can be easily speculated from the results in GNN[1].

[1]Neural Sheaf Diffusion: A Topological Perspective on Heterophily and Oversmoothing in GNNs.

[2]Hypergraph neural networks.

[3]Hypergcn: A new method for training graph convolutional networks on hypergraphs.

[4]Preventing over-smoothing for hypergraph neural networks

Minor:

Some references have incorrect author names. E.g. [43] Guanzi Chen, Jiying Zhang,et al. Preventing over-smoothing for hypergraph neural networks


**Questions:**

It is necessary to highlight the difficulties of using sheaf to hypergraphs and the core technological innovation of the proposed method.

**Limitations:**

Yes.

---

> ### Author Rebuttal · Authors · 2023-08-09
>
> We warmly appreciate your time and effort invested in reviewing our paper. Your perceptive comments and constructive criticism are invaluable, and we are grateful. We are sorry for the incorrect reference and we will make sure it will be fixed in the final version of the paper.
>
> **Regarding the novelty**
>
> We acknowledge your feedback on the novelty of our work. Due to their remarkable versatility and powerful representation capabilities, sheaves have become a growing field within the machine learning community. Yet, our work is pioneering in introducing the cellular sheaf *for hypergraphs*. This mathematical construct enriches hypergraphs by associating a vector space with each node and hyperedge, along with linear projections facilitating the transfer of information between them. The cellular sheaf brings a unique perspective and adds a fresh dimension to the current body of knowledge inside the hypergraph community. We emphasize the complexity of extending concepts from graphs to hypergraphs due to the latter's increased complexity. This endeavor in itself is a significant contribution to the field.
>
> Moving on to the Sheaf hypergraph networks, it's easy to spot similarities with the Sheaf Convolutional Network (SCN) in terms of form. Hypergraph represents a  generalisation of graphs, that’s why obtaining similar processing with the ones in the graph literature it’s natural when constraining the hypergraph to be a graph.  Nevertheless, the underlying mathematical structure and implementation diverge substantially. This is accounted for by the unique, higher-order nature of hypergraphs and their respective Laplacians.
>
> In the linear case of Sheaf hypergraph Laplacian, the sheaf Laplacian is defined as $\mathcal{L} x_v=\sum_{e; v \in e} F_{v  \trianglelefteq e}^T \sum_{u \in e} (F_{v  \trianglelefteq e}x_v - F_{u  \trianglelefteq e} x_u)$. When each hyperedge contains exactly two nodes (thus, $\mathcal{H}$ is a graph), the inner summation will have a single term, and we recover the sheaf Laplacian for graphs as formulated in [1]. However, the power of higher-order processing consists of being able to process relations beyond pairwise. We achieve this by allowing the sheaf to act on edges of arbitrary cardinality.
>
> Moreover, the sheaf Laplacian for the non-linear case differs substantially from [1]. In fact, when the sheaf is restricted to the trivial case ($d = 1$ and  $F_{v  \trianglelefteq e}= 1$), this corresponds to the non-linear sheaf Laplacian of a hypergraph, as introduced in [2]. Our work generalises [2] by incorporating sheaf Laplacian, thereby introducing a more diverse range of inductive biases and avoiding well-known challenges in the hypergraph community such as over-smoothing.
>
> **Similarity with hypergraph Dirichlet energy**
>
> The main goal of our theoretical results is not just to empirically compare the SheafHNN and HNN models but to highlight the benefits SheafHNN brings on top of HNN. We do not aim to introduce a different metric to evaluate hypergraph issues as in [3]. Instead, our goal is to emphasise the differences in the minimised energy between the two methods. A comparison of the energies minimized by SheafHNN and HNN reveals the distinctive capabilities of the two models: while HNN narrows the features of the nodes, making it susceptible to over-smoothing, SheafHNN optimizes the representation of the nodes in the sheaf space, effectively circumventing over-smoothing. In Figure 2, we also provide empirical validation by plotting the Dirichlet energy, which serves as a measure of over-smoothing [3], for both models. The results demonstrate that while the Dirichlet energy of HNN decreases as the number of layers increases, the Dirichlet energy of SheafHNN remains relatively constant.
>
> The results in [3] confine themselves to the linear case, and generalizing these results to the non-linear HGCN is a highly challenging task due to the non-differentiability of the non-linear operator. Our work, however, addresses both these aspects, providing a comprehensive and general overview and a more complete intuition over what advantages sheaf hypergraphs can bring to the hypergraph research area.
>
> Considering these, we humbly assert that our work is indeed novel and makes a significant contribution to the discourse in this field.
>
> *[1] Cristian Bodnar, Francesco Di Giovanni, Benjamin Paul Chamberlain, Pietro Liò, and Michael M. Bronstein. Neural sheaf diffusion: A topological perspective on heterophily and oversmoothing in GNNs.*
>
> *[2] Matthias Hein, Simon Setzer, Leonardo Jost, and Syama Sundar Rangapuram. The total variation on hypergraphs - learning on hypergraphs revisited.*
>
> *[3] Guanzi Chen, Jiying Zhang, Xi Xiao, Yang Li, Preventing Over-Smoothing for Hypergraph Neural Networks.*

---

> > ### Comment · Reviewer_C7Ao · 2023-08-15
> >
> > Thanks for the author's reply. The response addressed my questions. I still keep my rating.

---

### Decision · Program_Chairs · 2023-09-21

**Decision:**

Accept (poster)

**Comment:**

This paper introduces cellular sheaf for hypergraphs, which associates a vector space for each node and hyperedge, as well as a linear map between each incident node-hyperedge pair. The linear map is learned from the node-hyperedge features, aiming to project nodes to hyperedge-specific embedding space before performing the diffusion/message passing, thus enabling a higher expressive power compared to traditional hypergraph Laplacian that treats nodes equally within a hyperedge. The proposed methods achieve superior performance on multiple benchmark datasets. One improvement area is on the presentation: the motivation of cellular sheaf can be more clearly discussed. More explanation and intuition could be put to the introduction part, which will make the paper easier to understand for readers not familiar with cellular sheaf. Overall, an acceptance is recommended by the AC.